# On Pathologies in KL-Regularized Reinforcement Learning from Expert Demonstrations

**Tim G. J. Rudner**[*][†]
University of Oxford

**Cong Lu**[*]
University of Oxford

**Michael A. Osborne**
University of Oxford

**Yarin Gal**
University of Oxford

**Yee Whye Teh**
University of Oxford

## Abstract

KL-regularized reinforcement learning from expert demonstrations has proved successful in improving the sample efficiency of deep reinforcement learning algorithms, allowing them to be applied to challenging physical real-world tasks. However, we show that KL-regularized reinforcement learning with behavioral reference policies derived from expert demonstrations can suffer from pathological training dynamics that can lead to slow, unstable, and suboptimal online learning. We show empirically that the pathology occurs for commonly chosen behavioral policy classes and demonstrate its impact on sample efficiency and online policy performance. Finally, we show that the pathology can be remedied by *non-parametric* behavioral reference policies and that this allows KL-regularized reinforcement learning to significantly outperform state-of-the-art approaches on a variety of challenging locomotion and dexterous hand manipulation tasks.

## 1  Introduction

Reinforcement learning (RL) [15, 24, 46, 47] is a powerful paradigm for learning complex behaviors. Unfortunately, many modern reinforcement learning algorithms require agents to carry out millions of interactions with their environment to learn desirable behaviors, making them of limited use for a wide range of practical applications that cannot be simulated [8, 28]. This limitation has motivated the study of algorithms that can incorporate pre-collected offline data into the training process either fully offline or with online exploration to improve sample efficiency, performance, and reliability [2, 6, 16, 23, 52, 53]. An important and well-motivated subset of these methods consists of approaches for efficiently incorporating expert demonstrations into the learning process [5, 11, 18, 42].

Reinforcement learning with Kullback-Leibler (KL) regularization is a particularly successful approach for doing so [3, 27, 29, 31, 44, 51]. In KL-regularized reinforcement learning, the standard reinforcement learning objective is augmented by a Kullback-Leibler divergence term that penalizes dissimilarity between the online policy and a behavioral reference policy derived from expert demonstrations. The resulting regularized objective pulls the agent's online policy towards the behavioral reference policy while also allowing it to improve upon the behavioral reference policy by exploring and interacting with the environment. Recent advances that leverage explicit or implicit KL-regularized objectives, such as BRAC [51], ABM [44], and AWAC [27], have shown that KL-regularized reinforcement learning from expert demonstrations is able to significantly improve the sample efficiency of online training and reliably solve challenging environments previously unsolved by standard deep reinforcement learning algorithms.

---

[*]Equal contribution. [†] Corresponding author: `tim.rudner@cs.ox.ac.uk`.

35th Conference on Neural Information Processing Systems (NeurIPS 2021).

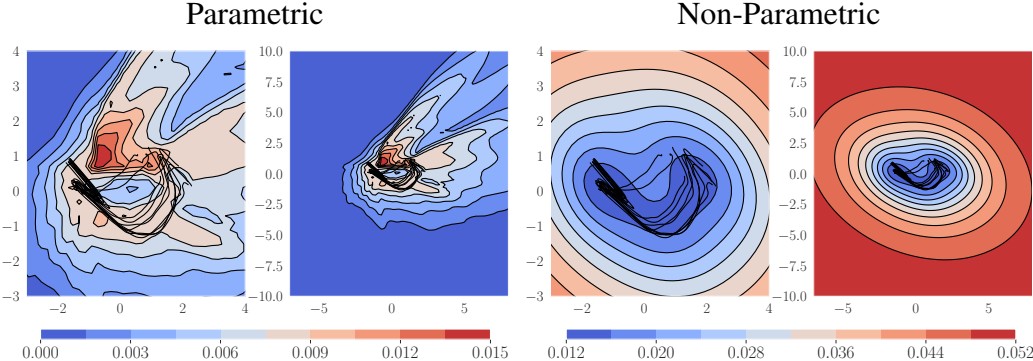

**Figure 1:** Predictive variances of non-parametric and parametric behavioral policies on a low dimensional representation (the first two principal components) of a 39-dimensional dexterous hand manipulation state space (see "door-binary-v0" in Figure 5). **Left**: Parametric neural network Gaussian behavioral policy $\pi_\psi(\cdot \,|\, \mathbf{s}) = \mathcal{N}(\boldsymbol{\mu}_\psi(\mathbf{s}), \boldsymbol{\sigma}_\psi^2(\mathbf{s}))$. **Right**: Non-parametric Gaussian process posterior behavioral policy $\pi_{\mathcal{GP}}(\cdot \,|\, \mathbf{s}, \mathcal{D}_0) = \mathcal{GP}(\boldsymbol{\mu}_0(\mathbf{s}), \boldsymbol{\Sigma}_0(\mathbf{s}, \mathbf{s}'))$. Expert trajectories $\mathcal{D}$ used to train the behavioral policies are shown in black. The GP predictive variance is well-calibrated: It is small near the expert trajectories and large in other parts of the state space. In contrast, the neural network predictive variance is poorly calibrated: It is relatively small on the expert trajectories, and collapses to near zero elsewhere. Note the significant difference in scales.

**Contributions.** In this paper, we show that despite some empirical success, KL-regularized reinforcement learning from expert demonstrations can suffer from previously unrecognized pathologies that lead to instability and sub-optimality in online learning. To summarize, our core contributions are as follows:

- We illustrate empirically that commonly used classes of parametric behavioral policies experience a collapse in predictive variance about states away from the expert demonstrations.

- We demonstrate theoretically and empirically that KL-regularized reinforcement learning algorithms can suffer from pathological training dynamics in online learning when regularized against behavioral policies that exhibit such a collapse in predictive variance.

- We show that the pathology can be remedied by *non-parametric* behavioral policies, whose predictive variances are well-calibrated and guaranteed not to collapse about previously unseen states, and that fixing the pathology results in online policies that significantly outperform state-of-the-art approaches on a range of challenging locomotion and dexterous hand manipulation tasks.

The left panel of Figure 1 shows an example of the collapse in predictive variance away from the expert trajectories in parametric behavioral policies. In contrast, the right panel of Figure 1 shows the predictive variance of a non-parametric behavioral policy, which—unlike in the case of the parametric policy—increases off the expert trajectories. By avoiding the pathology, we obtain a stable and reliable approach to sample-efficient reinforcement learning, applicable to a wide range of reinforcement learning algorithms that leverage KL-regularized objectives.[2]

## 2  Background

We consider the standard reinforcement learning setting where an agent interacts with a discounted Markov Decision Process (MDP) [46] given by a 5-tuple $(\mathcal{S}, \mathcal{A}, p, r, \gamma)$, where $\mathcal{S}$ and $\mathcal{A}$ are the state and action spaces, $p(\cdot \,|\, \mathbf{s}_t, \mathbf{a}_t)$ are the transition dynamics, $r(\mathbf{s}_t, \mathbf{a}_t)$ is the reward function, and $\gamma$ is a discount factor. $\rho_\pi(\boldsymbol{\tau}_t)$ denotes the state–action trajectory distribution from time $t$ induced by a policy $\pi(\cdot \,|\, \mathbf{s}_t)$. The discounted return from time step $t$ is given by $R(\boldsymbol{\tau}_t) = \sum_{k=t}^{\infty} \gamma^k r(\mathbf{s}_k, \mathbf{a}_k)$ for $t \in \mathbb{N}_0$. The standard reinforcement learning objective to be maximized is the expected discounted return $J_\pi(\boldsymbol{\tau}_0) = \mathbb{E}_{\rho_\pi(\boldsymbol{\tau}_0)}[R(\boldsymbol{\tau}_0)]$ under the policy trajectory distribution.

### 2.1  Improving and Accelerating Online Training via Behavioral Cloning

We consider settings where we have a set of expert demonstrations *without reward*, $\mathcal{D}_0 = \{(\mathbf{s}_n, \mathbf{a}_n)\}_{n=1}^{N} = \{\bar{\mathbf{S}}, \bar{\mathbf{A}}\}$, which we would like to use to speed up and improve online learn-

---

[2]Code and visualizations of our results can be found at https://sites.google.com/view/nppac.

ing [5, 42]. A standard approach for turning expert trajectories into a policy is behavioral cloning [1, 4] which involves learning a mapping from states in the expert demonstrations to their corresponding actions, that is, $\pi_0 : \mathcal{S} \to \mathcal{A}$. As such, behavioral cloning does not assume or require access to a reward function and only involves learning a mapping from states to action in a supervised fashion.

Since expert demonstrations are costly to obtain and often only available in small number, behavioral cloning alone is typically insufficient for agents to learn good policies in complex environments and has to be complemented by a method that enables the learner to build on the cloned behavior by interacting with the environment. A particularly successful and popular class of algorithms used for incorporating behavioral policies into online training is KL-regularized reinforcement learning [10, 37, 43, 48].

## 2.2 KL-Regularized Objectives in Reinforcement Learning

KL-regularized reinforcement learning modifies the standard reinforcement learning objective by augmenting the return with a negative KL divergence term from the learned policy $\pi$ to a reference policy $\pi_0$, given a temperature parameter $\alpha$. The resulting discounted return from time step $t \in \mathbb{N}_0$ is then given by

$$\tilde{R}(\boldsymbol{\tau}_t) = \sum_{k=t}^{\infty} \gamma^k \big[ r(\mathbf{s}_k, \mathbf{a}_k) - \alpha \mathbb{D}_{\text{KL}}(\pi(\cdot \,|\, \mathbf{s}_k) \,\|\, \pi_0(\cdot \,|\, \mathbf{s}_k)) \big] \tag{1}$$

and the reinforcement learning objective becomes $\tilde{J}_\pi(\boldsymbol{\tau}_0) = \mathbb{E}_{\rho_\pi(\boldsymbol{\tau}_0)}[\tilde{R}(\boldsymbol{\tau}_0)]$. When the reference policy $\pi_0$ is given by a uniform distribution, we recover the entropy-regularized reinforcement learning objective used in Soft Actor–Critic (SAC) [13] up to an additive constant.

Under a uniform reference policy $\pi_0$, the resulting objective encourages exploration, while also choosing high-reward actions. In contrast, when $\pi_0$ is non-uniform, the agent is discouraged to explore areas of the state space $\mathcal{S}$ where the variance of $\pi_0(\cdot \,|\, \mathbf{s})$ is low (i.e., more certain) and encouraged to explore areas of the state space where the variance of $\pi_0(\cdot \,|\, \mathbf{s})$ is high. The KL-regularized reinforcement learning objective can be optimized via policy–gradient and actor–critic algorithms.

## 2.3 KL-Regularized Actor–Critic

An optimal policy $\pi$ that maximizes the expected KL-augmented discounted return $\tilde{J}_\pi$ can be learned by directly optimizing the policy gradient $\nabla_\pi \tilde{J}_\pi$. However, this policy gradient estimator exhibits high variance, which can lead to unstable learning. Actor–critic algorithms [7, 17, 32, 38] attempt to reduce this variance by making use of the state value function $V^\pi(\mathbf{s}_t) = \mathbb{E}_{\rho_\pi(\boldsymbol{\tau}_t)}[\tilde{R}(\boldsymbol{\tau}_t) \,|\, \mathbf{s}_t]$ or the state–action value function $Q^\pi(\mathbf{s}_t, \mathbf{a}_t) = \mathbb{E}_{\rho_\pi(\boldsymbol{\tau}_t)}[\tilde{R}(\boldsymbol{\tau}_t) \,|\, \mathbf{s}_t, \mathbf{a}_t]$ to stabilize training.

Given a reference policy $\pi_0(\mathbf{a}_t \,|\, \mathbf{s}_t)$, the state value function can be shown to satisfy the modified Bellman equation

$$V^\pi(\mathbf{s}_t) \doteq \mathbb{E}_{\mathbf{a}_t \sim \pi(\cdot|\mathbf{s}_t)}[Q^\pi(\mathbf{s}_t, \mathbf{a}_t)] - \alpha \mathbb{D}_{\text{KL}}\big(\pi(\cdot \,|\, \mathbf{s}_t) \,\|\, \pi_0(\cdot \,|\, \mathbf{s}_t)\big)$$

with a recursively defined $Q$-function

$$Q^\pi(\mathbf{s}_t, \mathbf{a}_t) \doteq r(\mathbf{s}_t, \mathbf{a}_t) + \gamma \mathbb{E}_{\mathbf{s}_{t+1} \sim p(\cdot|\mathbf{s}_t, \mathbf{a}_t)}[V^\pi(\mathbf{s}_{t+1})].$$

Instead of directly optimizing the objective function $\tilde{J}_\pi$ via the policy gradient, actor–critic methods alternate between policy evaluation and policy improvement [7, 13]:

**Policy Evaluation.** During the policy evaluation step, $Q_\theta^\pi(\mathbf{s}, \mathbf{a})$, parameterized by parameters $\theta$, is trained by minimizing the Bellman residual

$$J_Q(\theta) \doteq \mathbb{E}_{(\mathbf{s}_t, \mathbf{a}_t) \sim \mathcal{D}}\Big[(Q_\theta(\mathbf{s}_t, \mathbf{a}_t) - (r(\mathbf{s}_t, \mathbf{a}_t) + \gamma \mathbb{E}_{\mathbf{s}_{t+1} \sim p(\cdot|\mathbf{s}_t, \mathbf{a}_t)}[V_{\bar{\theta}}(\mathbf{s}_{t+1})]))^2\Big], \tag{2}$$

where $\mathcal{D}$ is a replay buffer and $\bar{\theta}$ is a stabilizing moving average of parameters.

**Policy Improvement.** In the policy improvement step, the policy $\pi_\phi$, parameterized by parameters $\phi$, is updated towards the exponential of the KL-augmented $Q$-function,

$$J_\pi(\phi) \doteq \mathbb{E}_{\mathbf{s}_t \sim \mathcal{D}}\left[\alpha \mathbb{D}_{\text{KL}}(\pi_\phi(\cdot \,|\, \mathbf{s}_t) \,\|\, \pi_0(\cdot \,|\, \mathbf{s}_t))\right] - \mathbb{E}_{\mathbf{s}_t \sim \mathcal{D}}\left[\mathbb{E}_{\mathbf{a}_t \sim \pi_\phi(\cdot|\mathbf{s}_t)}\left[Q_\theta(\mathbf{s}_t, \mathbf{a}_t)\right]\right], \tag{3}$$

with states sampled from a replay buffer $\mathcal{D}$ and actions sampled from the parameterized online policy $\pi_\phi$. The following sections will focus on the policy improvement objective and how certain types of references policies can lead to pathologies when optimizing $J_\pi(\phi)$ with respect to $\phi$.

# 3 Identifying the Pathology

In this section, we investigate the effect of KL-regularization on the training dynamics. To do so, we first consider the properties of the KL divergence to identify a potential failure mode for KL-regularized reinforcement learning. Next, we consider parametric Gaussian behavioral reference policies commonly used in practice for continuous control tasks [13, 51] and show that for Gaussian behavioral reference policies with small predictive variance, the policy improvement objective suffers from exploding gradients with respect to the policy parameters $\phi$. We confirm that this failure occurs empirically and demonstrate that it results in slow, unstable, and suboptimal online learning. Lastly, we show that various regularization techniques used for estimating behavioral policies are unable to prevent this failure and also lead to suboptimal online policies.

## 3.1 When Are KL-Regularized Reinforcement Learning Objectives Meaningful?

We start by considering the properties of the KL divergence and discuss how these properties can lead to potential failure modes in KL-regularized objectives. A well-known property of KL-regularized objectives in the variational inference literature is the occurrence of singularities when the support of one distribution is not contained in the support of the other.

To illustrate this problem, we consider the case of Gaussian behavioral and online policies commonly used in practice. Mathematically, the KL divergence between two full Gaussian distributions is always finite and well-defined. Hence, we might hope KL-regularized reinforcement learning with Gaussian behavioral and online policies to be unaffected by the failure mode described above. However, the support of a Gaussian online policy $\pi_\phi(\cdot \,|\, \mathbf{s}_t)$ will not be contained in the support of a behavioral reference policy $\pi_0(\cdot \,|\, \mathbf{s}_t)$ as the predictive variance $\boldsymbol{\sigma}_0^2(\mathbf{s}_t)$ tends to zero, and hence $\mathbb{D}_{\mathrm{KL}}(\pi_\phi(\cdot \,|\, \mathbf{s}_t) \,\|\, \pi_0(\cdot \,|\, \mathbf{s}_t)) \to \infty$ as $\boldsymbol{\sigma}_0^2(\mathbf{s}_t) \to 0$. In other words, as the variance of a behavioral reference policy tends to zero and the behavioral distribution becomes degenerate, the KL divergence blows up to infinity [25]. While in practice, Gaussian behavioral policy would not operate in the limit of zero variance, the functional form of the KL divergence between (univariate) Gaussians,

$$\mathbb{D}_{\mathrm{KL}}(\pi_\phi(\cdot \,|\, \mathbf{s}_t) \,\|\, \pi_0(\cdot \,|\, \mathbf{s}_t)) \propto \log \frac{\boldsymbol{\sigma}_0(\mathbf{s}_t)}{\boldsymbol{\sigma}_\phi(\mathbf{s}_t)} + \frac{\boldsymbol{\sigma}_\phi^2(\mathbf{s}_t) + (\boldsymbol{\mu}_\phi(\mathbf{s}_t) - \boldsymbol{\mu}_0(\mathbf{s}_t))^2}{2\boldsymbol{\sigma}_0^2(\mathbf{s}_t)},$$

implies a continuous, quadratic increase in the magnitude of the divergence as $\boldsymbol{\sigma}_0(\mathbf{s}_t)$ decreases, further exacerbated by a large difference in predictive means, $|\boldsymbol{\mu}_\phi(\mathbf{s}_t) - \boldsymbol{\mu}_0(\mathbf{s}_t)|$.

As a result, for Gaussian behavioral reference policies $\pi_0(\cdot \,|\, \mathbf{s}_t)$ that assign very low probability to sets of points in sample space far away from the distribution's mean $\boldsymbol{\mu}_0(\mathbf{s}_t)$, computing the KL divergence can result in divergence values so large to cause numerical instabilities and arithmetic overflow. Hence, even for a suitably chosen behavioral reference policy class, vanishingly small behavioral reference policy predictive variances can cause the KL divergence to 'blow up' and cause numerical issues at evaluation points far away from states in the expert demonstrations.

One way to address this failure mode may be to lower-bound the output of the variance network (e.g., by adding a small constant bias). However, placing a floor on the predictive variance of the behavioral reference policy is not sufficient to encourage effective learning. While it would prevent the KL divergence from blowing up, it would also lead to poor gradient signals, as well-calibrated predictive variance estimates that *increase* on states far away from the expert trajectories are necessary to keep the KL penalty from pulling the predictive mean of the online policy towards poor behavioral reference policy predictive means on states off the expert trajectories. Another possible solution could be to use heavy-tailed behavioral reference policies distributions, for example, Laplace distributions, to avoid pathological training dynamics. However, in Appendix B.3 we show that Laplace behavioral reference policies also suffer from pathological training dynamics, albeit less severely.

In the following sections, we explain how an explosion in $\mathbb{D}_{\mathrm{KL}}(\pi_\phi(\cdot \,|\, \mathbf{s}_t) \,\|\, \pi_0(\cdot \,|\, \mathbf{s}_t))$ caused by small $\boldsymbol{\sigma}_0^2(\mathbf{s}_t)$ affects the gradients of $J_\pi(\phi)$ in KL-regularized RL and discuss of how and why $\boldsymbol{\sigma}_0^2(\mathbf{s}_t)$ may tend to zero in practice.

## 3.2 Exploding Gradients in KL-Regularized Reinforcement Learning Objectives

To understand how small predictive variances in behavioral reference policies can affect—and possibly destabilize—online training in KL-regularized RL, we consider the contribution of the behavioral

reference policy's variance to the gradient of the policy objective in Equation (3). Compared to entropy-regularized actor–critic methods (SAC, Haarnoja et al. [13]), which implicitly regularize against a uniform policy, the gradient estimator $\hat{\nabla}_\phi J_\pi(\phi)$ in KL-regularized RL gains an extra scaling term $\nabla_{\mathbf{a}_t} \log \pi_0(\mathbf{a}_t \,|\, \mathbf{s}_t)$, the gradient of the prior log-density evaluated actions $\mathbf{a}_t \sim \pi_\phi(\cdot \,|\, \mathbf{s})$:

**Proposition 1** (Exploding Gradients in KL-Regularized RL). *Let $\pi_0(\cdot \,|\, \mathbf{s})$ be a Gaussian behavioral reference policy with mean $\boldsymbol{\mu}_0(\mathbf{s}_t)$ and variance $\boldsymbol{\sigma}_0^2(\mathbf{s}_t)$, and let $\pi_\phi(\cdot \,|\, \mathbf{s})$ be an online policy with reparameterization $\mathbf{a}_t = f_\phi(\epsilon_t; \mathbf{s}_t)$ and random vector $\epsilon_t$. The gradient of the policy loss with respect to the online policy's parameters $\phi$ is then given by*

$$\hat{\nabla}_\phi J_\pi(\phi) = \big(\alpha \nabla_{\mathbf{a}_t} \log \pi_\phi(\mathbf{a}_t \,|\, \mathbf{s}_t) - \alpha \nabla_{\mathbf{a}_t} \log \pi_0(\mathbf{a}_t \,|\, \mathbf{s}_t) \tag{4}$$
$$- \nabla_{\mathbf{a}_t} Q(\mathbf{s}_t, \mathbf{a}_t)\big) \nabla_\phi f_\phi(\epsilon_t; \mathbf{s}_t) + \alpha \nabla_\phi \log \pi_\phi(\mathbf{a}_t \,|\, \mathbf{s}_t)$$

*with* $\nabla_{\mathbf{a}_t} \log \pi_0(\mathbf{a}_t \,|\, \mathbf{s}_t) = -\frac{\mathbf{a}_t - \boldsymbol{\mu}_0(\mathbf{s}_t)}{\boldsymbol{\sigma}_0^2(\mathbf{s}_t)}$. *For fixed* $|\mathbf{a}_t - \boldsymbol{\mu}_0(\mathbf{s}_t)|$, $\nabla_{\mathbf{a}_t} \log \pi_0(\mathbf{a}_t | \mathbf{s}_t)$ *grows as* $\mathcal{O}(\boldsymbol{\sigma}_0^{-2}(\mathbf{s}_t))$; *thus,*

$$|\hat{\nabla}_\phi J_\pi(\phi)| \to \infty \quad as \quad \boldsymbol{\sigma}_0^2(\mathbf{s}_t) \to 0 \quad whenever \quad \nabla_\phi f_\phi(\epsilon_t; \mathbf{s}_t) \neq 0.$$

*Proof.* See Appendix A.1. □

This result formalizes the intuition presented in Section 3.1 that a behavioral reference policy with a sufficiently small predictive variance may cause KL-regularized reinforcement learning to suffer from pathological training dynamics in gradient-based optimization. The smaller the behavioral reference policy's predictive variance, the more sensitive the policy objective's gradients will be to differences in the means of the online and behavioral reference policies. As a result, for behavioral reference policies with small predictive variance, the KL divergence will heavily penalize online policies whose predictive means diverge from the predictive means of the behavioral policy—even in regions of the state space away from the expert trajectory where the behavioral policy's mean prediction is poor.

### 3.3 Predictive Uncertainty Collapse Under Parametric Policies

The most commonly used method for estimating behavioral policies is maximum likelihood estimation (MLE) [44, 51], where we seek $\pi_0 \doteq \pi_{\psi^\star}$ with $\psi^\star \doteq \arg\max_\psi \{\mathbb{E}_{(\mathbf{s}, \mathbf{a}) \sim \mathcal{D}_0}[\log \pi_\psi(\mathbf{a} \,|\, \mathbf{s})]\}$ for a parametric behavioral policy $\pi_\psi$. In practice, $\pi_\psi$ is often assumed to be Gaussian, $\pi_\psi(\cdot \,|\, \mathbf{s}) = \mathcal{N}(\boldsymbol{\mu}_\psi(\mathbf{s}), \boldsymbol{\sigma}_\psi^2(\mathbf{s}))$, with $\boldsymbol{\mu}_\psi(\mathbf{s})$ and $\boldsymbol{\sigma}_\psi^2(\mathbf{s})$ parameterized by a neural network.

While maximizing the likelihood of the expert trajectories under the behavioral policy is a sensible choice for behavioral cloning, the limited capacity of the neural network parameterization can produce unwanted behaviors in the resulting policy. The maximum likelihood objective ensures that the behavioral policy's predictive mean reflects the expert's actions and the predictive variance the (aleatoric) uncertainty inherent in the expert trajectories.

However, the maximum likelihood objective encourages parametric policies to use their model capacity toward fitting the expert demonstrations and reflecting the aleatoric uncertainty in the data. As a result, for states off the expert trajectories, the policy can become degenerate and collapse to point predictions instead of providing meaningful predictive variance estimates that reflect that the behavioral policy ought to be highly uncertain about its predictions in previously unseen regions of the state space. Similar behaviors are well-known in parametric probabilistic models and well-documented in the approximate Bayesian inference literature [33, 39].

Figure 1 demonstrates the collapse in predictive variance under maximum likelihood estimation in a low-dimensional representation of the "door-binary-v0" dexterous hand manipulation environment. It shows that while the predictive variance is small close to the expert trajectories (depicted as black lines), it rapidly decreases further away from them. Examples of variance collapse in other environments are presented in Appendix B.6. Figure 2 shows that the predictive variance off the expert trajectories consis-

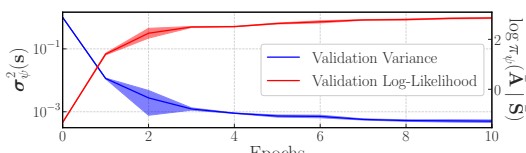

**Figure 2:** Collapse in the predictive variance (in **blue**) of a Gaussian behavioral policy parameterized by a neural network when training via maximum likelihood estimation. Lines and shaded regions denote means and standard deviations over five random seeds, respectively.

tently decreases during training. As shown in Proposition 1, such a collapse in predictive variance can

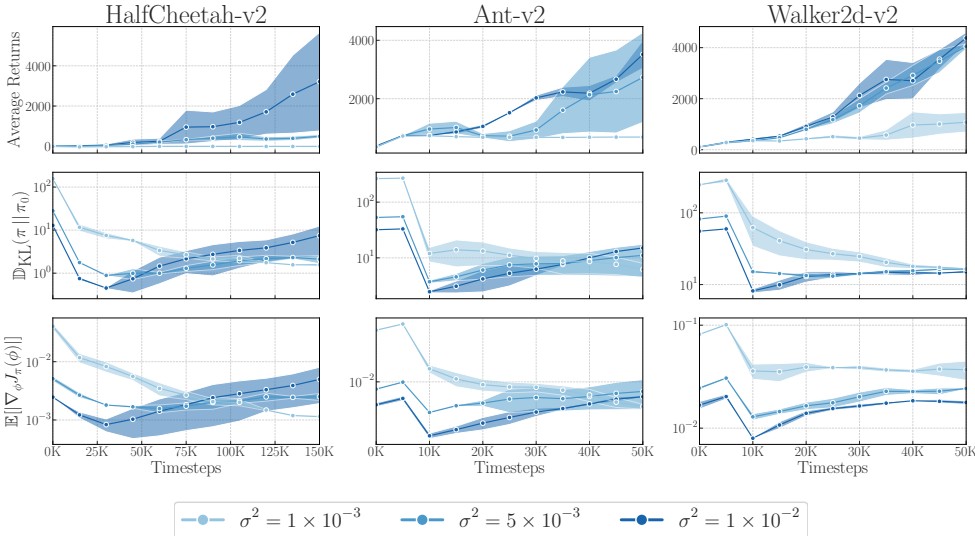

**Figure 3:** Ablation study showing the effect of predictive variance collapse on the performance of KL-regularized RL on MuJoCo environments. The plots show the average return of the learned policy, the magnitude of the KL penalty, and the magnitude of the average absolute gradients of the policy loss during online training. The lighter the shading, the lower the behavioral policy's predictive variance.

result in pathological training dynamics in KL-regularized online learning—steering the online policy towards suboptimal trajectories in regions of the state space far away from the expert demonstrations and deteriorating performance.

**Effect of regularization on uncertainty collapse.** To prevent a collapse in the behavioral policy's predictive variance, prior work proposed adding entropy or Tikhonov regularization to the MLE objective [51]. However, doing so does not succeed in preventing a collapse in predictive variance off the expert demonstration trajectories, as we show in Appendix A.3. Deep ensembles [20], whose predictive mean and variance are computed from the predictive means and variances of multiple Gaussian neural networks, are a widely used method for uncertainty quantification in regression settings. However, model ensembling can be costly and unreliable, as it requires training multiple neural networks from scratch and does not guarantee well-calibrated uncertainty estimates [39, 49]. We provide visualizations in Appendix B.5 which show that ensembling multiple neural network policies does not fully prevent a collapse in predictive variance.

### 3.4 Empirical Confirmation of Uncertainty Collapse

To confirm Proposition 1 empirically and assess the effect of the collapse in predictive variance on the performance of KL-regularized RL, we perform an ablation study where we fix the predictive mean function of a behavioral policy to a mean function that attains 60% of the optimal performance and vary the magnitude of the policy's predictive variance. Specifically, we set the behavioral policy's predictive variance to different constant values in the set $\{1 \times 10^{-3}, 5 \times 10^{-3}, 1 \times 10^{-2}\}$ (following a similar implementation in Nair et al. [27]).[3] The results of this experiment are shown in Figure 3, which shows the average returns, the KL divergence, and the average absolute gradients of the policy loss over training. The plots confirm that as the predictive variance of the offline behavioral policy tends to zero, the KL terms and average policy gradient magnitude explode as implied by Proposition 1, leading to unstable training and a collapse or dampening in average returns.

In other words, even for behavioral policies with accurate predictive means, smaller predictive variances slow down or even entirely prevent learning good behavioral policies. This observation confirms that the pathology identified in Proposition 1 occurs in practice and that it can have a significant impact on KL-regularized RL from expert demonstrations, calling into question the usefulness of KL regularization as a means for accelerating and improving online training. In Appendix B.1, we show that an analogous relationship exists for the gradients of the $Q$-function loss.

---

[3] We attempted to use smaller values, but the gradients grew too large and caused arithmetic overflow.

## 4 Fixing the Pathology

In order to address the collapse in predictive uncertainty for behavioral policies parameterized by a neural network trained via MLE, we specify a *non-parametric* behavioral policy whose predictive variance is *guaranteed* not to collapse about previously unseen states. Noting that KL-regularized RL with a behavioral policy can be viewed as approximate Bayesian inference with an empirical prior policy [13, 21, 40], we propose *Non-Parametric Prior Actor–Critic* (N-PPAC), an off-policy temporal difference algorithm for improved, accelerated, and stable online learning with behavioral policies.

### 4.1 Non-Parametric Gaussian Processes Behavioral Policies

Gaussian processes (GPs) [36] are models over functions defined by a mean $m(\cdot)$ and covariance function $k(\cdot, \cdot)$. When defined in terms of a non-parametric covariance function, that is, a covariance function constructed from infinitely many basis functions, we obtain a non-degenerate GP, which has sufficient capacity to prevent a collapse in predictive uncertainty away from the training data. Unlike parametric models, whose capacity is limited by their parameterization, a non-parametric model's capacity *increases* with the amount of training data.

Considering a non-parametric GP behavioral policy, $\pi_0(\cdot \mid \mathbf{s})$, with

$$\mathbf{A} \mid \mathbf{s} \sim \pi_0(\cdot \mid \mathbf{s}) = \mathcal{GP}\big(m(\mathbf{s}), k(\mathbf{s}, \mathbf{s}')\big), \tag{5}$$

we can obtain a *non-degenerate* posterior distribution over actions conditioned on the offline data $\mathcal{D}_0 = \{\bar{\mathbf{S}}, \bar{\mathbf{A}}\}$ with actions sampled according to the

$$\mathbf{A} \mid \mathbf{s}, \mathcal{D}_0 \sim \pi_0(\cdot \mid \mathbf{s}, \mathcal{D}_0) = \mathcal{GP}\big(\boldsymbol{\mu}_0(\mathbf{s}), \boldsymbol{\Sigma}_0(\mathbf{s}, \mathbf{s}')\big), \tag{6}$$

with

$$\mu(\mathbf{s}) = m(\mathbf{s}) + k(\mathbf{s}, \bar{\mathbf{S}}) k(\bar{\mathbf{S}}, \bar{\mathbf{S}})^{-1} (\bar{\mathbf{A}} - m(\bar{\mathbf{A}})) \text{ and } \Sigma(\mathbf{s}, \mathbf{s}') = k(\mathbf{s}, \mathbf{s}') + k(\mathbf{s}, \bar{\mathbf{S}}) k(\bar{\mathbf{S}}, \bar{\mathbf{S}})^{-1} k(\bar{\mathbf{S}}, \mathbf{s}').$$

To obtain this posterior distribution, we perform exact Bayesian inference, which naively scales as $\mathcal{O}(N^3)$ in the number of training points $N$, but Wang et al. [50] show that exact inference in GP regression can be scaled to $N > 1,000,000$. Since expert demonstrations usually contain less than 100k datapoints, non-parametric GP behavioral policies are applicable to a wide array of real-world tasks. For an empirical evaluation of the time complexity of using a GP prior, see Section 5.5.

Figure 1 confirms that the non-parametric GP's predictive variance is well-calibrated: It is small in magnitude in regions of the state space near the expert trajectories and large in magnitude in other regions of the state space. While actor–critic algorithms like SAC implicitly use a uniform prior to explore the state space, using a behavioral policy with a well-calibrated predictive variance has the benefit that in regions of the state space close to the expert demonstrations the online policy learns to match the expert, while elsewhere the predictive variance increases and encourages exploration.

**Algorithmic details.** In our experiments, we use a KL-regularized objective with a standard actor–critic implementation and Double DQN [14]. Pseudocode is provided in (Appendix C.1).

## 5 Empirical Evaluation

We carry out a comparative empirical evaluation of our proposed approach vis-à-vis related methods that integrate offline data into online training. We provide a detailed description of the algorithms we compare against in Appendix A.4. We perform experiments on the MuJoCo benchmark suite and the substantially more challenging dexterous hand manipulation suite with sparse rewards.

We show that KL-regularized RL with a non-parametric behavioral reference policy can rapidly learn to solve difficult high-dimensional continuous control problems given only a small set of expert demonstrations and (often significantly) outperforms state-of-the-art methods, including ones that use offline reward information—which our approach does not require. Furthermore, we demonstrate that the GP behavioral policy's predictive variance is crucial for KL-regularized objectives to learn good online policies from expert demonstrations. Finally, we perform ablation studies that illustrate that non-parametric GP behavioral reference policies also outperform parametric behavioral reference policies with improved uncertainty quantification, such as deep ensembles and Bayesian neural

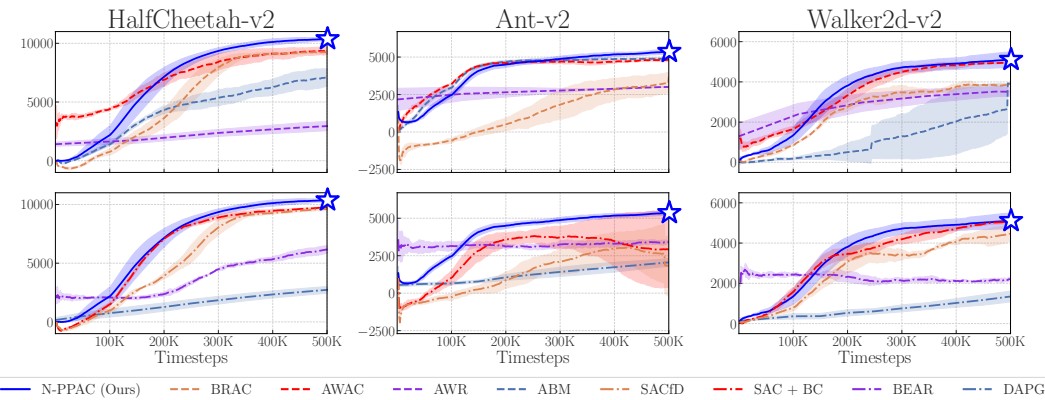

**Figure 4:** Comparison of N-PPAC (ours) vs. previous baselines on standard MuJoCo benchmark tasks. **Top**: KL-based methods (dashed lines), **Bottom**: Non-KL-based methods (dash-dotted lines). Both top and bottom plots include N-PPAC (**blue**). BRAC uses the same actor–critic algorithm as N-PPAC, but uses a parametric behavioral policy, and results in slower learning and worse final performance.

networks (BNNs) with Monte Carlo dropout, and that the difference between non-parametric and parametric models is exacerbated the fewer expert demonstrations are available. We use the expert data from Nair et al. [27], every experiment uses six random seeds, and we use a fixed KL-temperature for each environment class. For further implementation details, see Appendix C.2.

### 5.1 Environments

**MuJoCo locomotion tasks.** We evaluate N-PPAC on three representative tasks: "Ant-v2", "HalfCheetah-v2", and "Walker2d-v2". For each task, we use 15 demonstration trajectories collected by a pre-trained expert, each containing 1,000 steps. The behavioral policy is specified as the posterior distribution of a GP with a squared exponential kernel, which is well-suited for modeling smooth functions.

**Dexterous hand manipulation tasks.** Real-world robot learning is a setting where human demonstration data is readily available, and many deep RL approaches fail to learn efficiently. We study this setting in a suite of challenging dexterous manipulation tasks [35] using a 28-DoF five-fingered simulated ADROIT hand. The tasks simulate challenges common to real-world settings with high-dimensional action spaces, complex physics, and a large number of intermittent contact forces. We consider two tasks in particular: in-hand rotation of a pen to match a target and opening a door by unlatching and pulling a handle. We use binary rewards for task completion, which is significantly more challenging than the original setting considered in Rajeswaran et al. [35]. 25 expert demonstrations were provided for each task, each consisting of 200 environment steps which are not fully optimal but do successfully solve the task. The behavioral policy is specified as the posterior distribution of a GP with a Matérn kernel, which is more suitable for modeling non-smooth data.

### 5.2 Results

On MuJoCo environments, KL-regularized RL with a non-parametric behavioral policy consistently outperforms all related methods across all three tasks, successfully accelerating learning from offline data, as shown in Figure 4. Most notably, it outperforms methods such as AWAC [27]—the previous state-of-the-art—which attempts to eschew the problem of learning behavioral policies but instead uses an implicit constraint. Our approach, N-PPAC, exhibits an increase in stability and higher returns compared to comparable methods such as ABM and BRAC that explicitly regularize the online policy against a parametric behavioral policy and plateau at suboptimal performance levels as they are being forced to copy poor actions from the behavioral policy away from the expert data. In contrast, using a non-parametric behavioral policy allows us to avoid such undesirable behavior.

On dexterous hand manipulation environments, KL-regularized RL with a non-parametric behavioral policy performs on par or outperforms all related methods on both tasks, as shown in Figure 5. Most notably, on the door opening task, it achieves a stable success rate of 90% within only 100,000 environment interactions For comparison, AWAC requires 4× as many environment interactions to

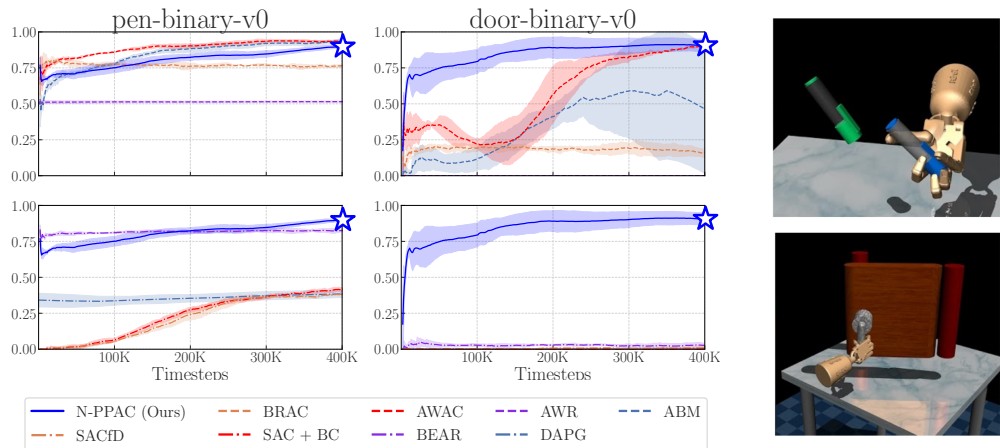

**Figure 5: Left & Center**: Comparison of N-PPAC (ours) vs. previous baselines on dexterous hand manipulation tasks. **Top**: KL-based methods (dashes), **Bottom**: Non-KL-based methods (dots and dashes). Both top and bottom plots include N-PPAC (**blue**). **Right**: The pen-binary-v0 (top) and door-binary-v0 (bottom) environments.

achieve the same performance and is significantly less stable, while most other methods fail to learn any meaningful behaviors.

**Alternative divergence metrics underperform KL-regularization.** KL-regularized RL with a non-parametric behavioral policy consistently outperforms methods that use alternative divergence metrics, as shown in the bottom plots of Figures 4 and 5.

## 5.3 Can the Pathology Be Fixed by Improved Parametric Uncertainty Quantification?

To assess whether the success of non-parametric behavioral reference policies is due to their predictive variance estimates—as suggested by Proposition 1—or due to better generalization from their predictive means, we perform an ablation study on the predictive variance of the behavioral policy. To isolate the effect of the predictive variance on optimization, we perform online training using behavioral policies with different predictive variance functions (parametric and non-parametric) and identical mean functions, which we set to be the predictive mean of the GP posterior (which achieves a success rate of ~80%). If the pathology identified in Proposition 1 can be remedied by commonly used parametric uncertainty quantification methods, we would expect the parametric and non-parametric behavioral policy variance functions to result in similar online policy success rates. We consider the challenging "door-binary-v0" environment for this ablation study.

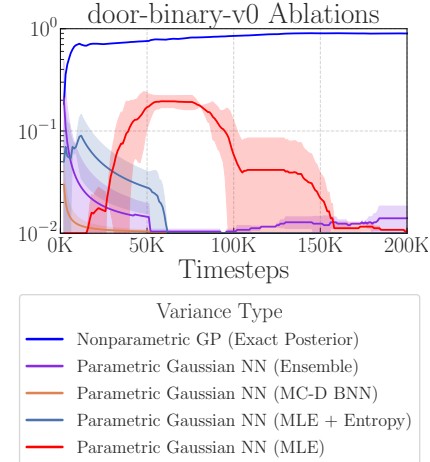

**Figure 6:** Post-online training success rates with different behavioral policy variance functions.

**Parametric uncertainty quantification is insufficient.** Figure 5 shows that parametric variance functions result in online policies that only achieve success rates of up to 20% and eventually deteriorate, whereas the non-parametric variance yields an online policy that achieves a success rate of nearly 100%. This finding shows that commonly used uncertainty quantification methods, such as deep ensembles or BNNs with Monte Carlo dropout, do not generate sufficiently well-calibrated uncertainty estimates to remedy the pathology, and better methods may be needed [9, 39, 41].

**Lower-bounding the predictive variance does not remedy the pathology.** The predictive variance of all MLE-based and ensemble behavioral reference policies in all experiments are bounded away from zero at a minimum value of $\approx 10^{-2}$. Hence, setting a floor on the variance is not sufficient to prevent pathological training dynamics. This result further demonstrates the importance of accurate predictive variance estimation in allowing the online policy to match expert actions in regions of the state space with low behavioral policy predictive variance and explore elsewhere.

### 5.4 Can a Single Expert Demonstration Be Sufficient to Accelerate Online Training?

To assess the usefulness of non-parametric behavioral reference policies in settings where only few expert demonstrations are available, we investigate whether the difference in performance between online policies trained with non-parametric and parametric behavioral reference policies, respectively, is exacerbated the fewer expert demonstrations are available. To answer this question, we consider the "HalfCheetah-v2" environment and compare online policies trained with different behavioral reference

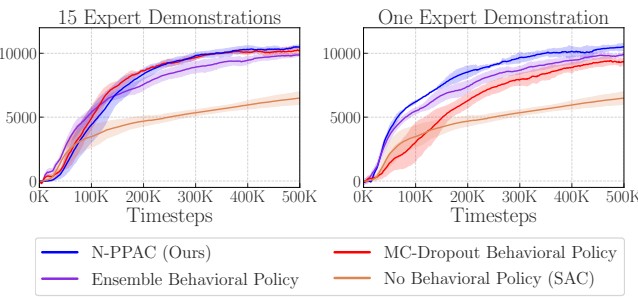

**Figure 7:** Returns during online training with different behavioral policies and varying amounts of expert demonstration data on "HalfCheetah-v2".

policies—non-parametric GPs, deep ensembles, and BNNs with Monte Carlo dropout—estimated either from 15 expert demonstrations (i.e., 15 state–action trajectories, containing 15,000 samples) or from a single expert demonstration (i.e., a single state–action trajectory, containing 1,000 samples).

**A single expert demonstration is sufficient for non-parametric behavioral reference policies.** Figure 7 shows the returns for online policies trained with behavioral reference policies estimated from the full dataset (top plot) and from only a single expert state–action trajectory (bottom plot). On the full dataset, we find that all three methods are competitive and improve on the prior state-of-the-art but that the GP behavioral policy leads to the highest return. Remarkably, non-parametric GP behavioral policies perform just as well with only a single expert demonstration as with all 15 (i.e., with 1,000 data points, instead of 15,000 data points). These results further emphasizes the usefulness of non-parametric behavioral policies when accelerating online training with expert demonstrations—even when only very few expert demonstrations are available.

### 5.5 Are Non-Parametric GP Behavioral Reference Policies Too Computationally Expensive?

Table 1 presents the time complexity of KL-regularized RL under non-parametric GP and parametric neural network behavioral reference policies, as measured by the average time elapsed per epoch on the "door-binary-v0" and "HalfCheetah-v2" environments. One epoch of online training on "door-binary-v0" and "HalfCheetah-v2" requires computing the KL divergence over 1,000 mini-batches of size 256 and 1,024, respectively. The time complexity of evaluating the log-density of a GP behavioral reference policy—needed for computing gradients of the KL divergence during online training—scales quadratically in the number of training data points and linearly in the dimensionality of the state and action space, respectively. As can be seen in Table 1, non-parametric GP behavioral reference policies only lead to a modest increase in the time needed to complete one epoch of training while resulting in significantly improved performance as shown in Figures 4 and 5.

**Table 1:** Time per epoch under different behavioral reference policies for expert demonstration data of varying size computed on a GeForce RTX 3080 GPU. The first and second value in each entry of the table give the time required when using a single parametric neural network and a GP behavioral reference policy, respectively.

| Dataset | 1,000 Data Points | 5,000 Data Points | 15,000 Data Points |
|---|---|---|---|
| HalfCheetah-v2 | 12.00s / 16.06s | 11.59s / 18.31s | 12.00s / 46.54s |
| door-binary-v0 | 19.62s / 23.78s | 19.62s / 33.62s | - |

## 6 Conclusion

We identified a previously unrecognized pathology in KL-regularized RL from expert demonstrations and showed that this pathology can significantly impede and even entirely prevent online learning. To remedy the pathology, we proposed the use of non-parametric behavioral reference policies, which we showed can significantly accelerate and improve online learning and yield online policies that (often significantly) outperform current state-of-the-art methods on challenging continuous control tasks. We hope that this work will encourage further research into better model classes for deep reinforcement learning algorithms, including and especially for reinforcement from image inputs.

## Acknowledgments and Disclosure of Funding

We thank Ashvin Nair for sharing his code and results, as well as for providing helpful insights about the dexterous hand manipulation suite. We also thank Clare Lyle, Charline Le Lan, and Angelos Filos for detailed feedback on an early draft of this paper, Avi Singh for early discussions about behavioral cloning in entropy-regularized RL, and Tim Pearce for a useful discussion on the role of good models in RL. TGJR and CL are funded by the Engineering and Physical Sciences Research Council (EPSRC). TGJR is also funded by the Rhodes Trust and by a Qualcomm Innovation Fellowship. We gratefully acknowledge donations of computing resources by the Alan Turing Institute.

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
