# Supplementary Material

## Table of Contents

## Appendix A  Derivations and Further Technical Details

### A.1  Proof of Proposition 1

**Proposition 1** (Exploding Gradients in KL-Regularized RL). *Let $\pi_0(\cdot \,|\, \mathbf{s})$ be a Gaussian behavioral reference policy with mean $\boldsymbol{\mu}_0(\mathbf{s})$ and variance $\boldsymbol{\sigma}_0^2(\mathbf{s})$, and let $\pi(\cdot \,|\, \mathbf{s})$ be an online policy with reparameterization $\mathbf{a}_t = f_\phi(\epsilon_t; \mathbf{s}_t)$ and random vector $\epsilon_t$. The gradient of the policy loss with respect to the online policy's parameters $\phi$ is then given by*

$$\hat{\nabla}_\phi J_\pi(\phi) = \big(\alpha \nabla_{\mathbf{a}_t} \log \pi_\phi(\mathbf{a}_t \,|\, \mathbf{s}_t) - \alpha \nabla_{\mathbf{a}_t} \log \pi_0(\mathbf{a}_t \,|\, \mathbf{s}_t) \\ - \nabla_{\mathbf{a}_t} Q(\mathbf{s}_t, \mathbf{a}_t)\big) \nabla_\phi f_\phi(\epsilon_t; \mathbf{s}_t) + \alpha \nabla_\phi \log \pi_\phi(\mathbf{a}_t \,|\, \mathbf{s}_t) \tag{A.1}$$

*with*

$$\nabla_{\mathbf{a}_t} \log \pi_0(\mathbf{a}_t \,|\, \mathbf{s}_t) = -\frac{\mathbf{a}_t - \boldsymbol{\mu}_0(\mathbf{s}_t)}{\boldsymbol{\sigma}_0^2(\mathbf{s}_t)}. \tag{A.2}$$

*For fixed $|\mathbf{a}_t - \boldsymbol{\mu}_0(\mathbf{s}_t)|$, $\nabla_{\mathbf{a}_t} \log \pi_0(\mathbf{a}_t \,|\, \mathbf{s}_t)$ grows as $\mathcal{O}(\boldsymbol{\sigma}_0^{-2}(\mathbf{s}_t))$; thus,*

$$|\hat{\nabla}_\phi J_\pi(\phi)| \to \infty \quad as \quad \boldsymbol{\sigma}_0^2(\mathbf{s}_t) \to 0, \tag{A.3}$$

*when $\nabla_\phi f_\phi(\epsilon_t; \mathbf{s}_t) \neq 0$.*

*Proof.* The policy loss, as given in Equation (3), is:

$$J_\pi(\phi) = \mathbb{E}_{\mathbf{s}_t \sim \mathcal{D}} \big[ \mathbb{D}_{\mathrm{KL}}\big(\pi_\phi(\cdot \,|\, \mathbf{s}_t) \,||\, \pi_0(\cdot \,|\, \mathbf{s}_t)\big) \big] - \mathbb{E}_{\mathbf{s}_t \sim \mathcal{D}} \big[ \mathbb{E}_{\mathbf{a}_t \sim \pi_\phi} [Q_\theta(\mathbf{s}_t, \mathbf{a}_t)] \big]. \tag{A.4}$$

To obtain a lower-variance gradient estimator, the policy is reparameterized using a neural network transformation

$$\mathbf{a}_t = f_\phi(\epsilon_t; \mathbf{s}_t) \tag{A.5}$$

where $\epsilon_t$ is an input noise vector. Following Haarnoja et al. [13], we can now rewrite Equation (A.4) as

$$J_\pi(\phi) = \mathbb{E}_{\mathbf{s}_t \sim \mathcal{D}, \epsilon_t} \left[ \alpha \big( \log \pi_\phi(f_\phi(\epsilon_t; \mathbf{s}_t) \mid \mathbf{s}_t) - \log \pi_0(f_\phi(\epsilon_t; \mathbf{s}_t) \mid \mathbf{s}_t) \big) - Q(\mathbf{s}_t, f_\phi(\epsilon_t; \mathbf{s}_t)) \right] \quad \text{(A.6)}$$

where $\mathcal{D}$ is a replay buffer and $\pi_\phi$ is defined implicitly in terms of $f_\phi$. We can approximate the gradient of Equation (A.6) with

$$\begin{aligned}
\hat{\nabla}_\phi J_\pi(\phi) = \big( \alpha \nabla_{\mathbf{a}_t} \log \pi_\phi(\mathbf{a}_t \mid \mathbf{s}_t) - \alpha \nabla_{\mathbf{a}_t} \log \pi_0(\mathbf{a}_t \mid \mathbf{s}_t) \\
- \nabla_{\mathbf{a}_t} Q(\mathbf{s}_t, \mathbf{a}_t) \big) \nabla_\phi f_\phi(\epsilon_t; \mathbf{s}_t) + \alpha \nabla_\phi \log \pi_\phi(\mathbf{a}_t \mid \mathbf{s}_t).
\end{aligned} \quad \text{(A.7)}$$

Next, consider the term $\nabla_{\mathbf{a}_t} \log \pi_0(\mathbf{a}_t \mid \mathbf{s}_t)$ for a Gaussian policy:

$$\log \pi_0(\mathbf{a}_t \mid \mathbf{s}_t) = \log \left( \frac{1}{\boldsymbol{\sigma}_0(\mathbf{s}_t)\sqrt{2\pi}} \right) - \frac{1}{2} \left( \frac{\mathbf{a}_t - \boldsymbol{\mu}_0(\mathbf{s}_t)}{\boldsymbol{\sigma}_0(\mathbf{s}_t)(\mathbf{s}_t)} \right)^2 \quad \text{(A.8)}$$

Thus,

$$\nabla_{\mathbf{a}_t} \log \pi_0(\mathbf{a}_t \mid \mathbf{s}_t) = -\frac{\mathbf{a}_t - \boldsymbol{\mu}_0(\mathbf{s}_t)}{\boldsymbol{\sigma}_0^2(\mathbf{s}_t)}. \quad \text{(A.9)}$$

For fixed $|\mathbf{a}_t - \boldsymbol{\mu}_0(\mathbf{s}_t)|$, $\nabla_{\mathbf{a}_t} \log(\pi_0(\mathbf{a}_t \mid \mathbf{s}_t))$ grows as $\mathcal{O}(\boldsymbol{\sigma}_0^{-2}(\mathbf{s}_t))$, and so,

$$|\hat{\nabla}_\phi J_\pi(\phi)| \to \infty \quad \text{as} \quad \boldsymbol{\sigma}_0^2(\mathbf{s}_t) \to 0. \quad \text{(A.10)}$$

whenever $\nabla_\phi f_\phi(\epsilon_t; \mathbf{s}_t) \neq 0$. $\qquad\square$

## A.2 Laplace Parametric Behavioral Reference Policy

A Laplace behavioral reference policy may be able to mitigate some of the problems posed by Proposition 1 due to the heavy tails of the distribution. The gradient for a Laplace behavioral reference policy

$$\pi_0(\mathbf{a}_t \mid \mathbf{s}_t) \doteq \frac{1}{2\boldsymbol{\sigma}_0(\mathbf{s}_t)} \exp \left( -\frac{|\mathbf{a}_t - \boldsymbol{\mu}_0(\mathbf{s}_t)|}{\boldsymbol{\sigma}_0(\mathbf{s}_t)} \right), \quad \text{(A.11)}$$

increases linearly for a given distance between $\mathbf{a}_t$ and the mean $\boldsymbol{\mu}_0(\mathbf{s}_t)$ as the scale $\boldsymbol{\sigma}_0(\mathbf{s}_t)$ tends to zero.

## A.3 Regularized Maximum Likelihood Estimation

To address the collapse in predictive variance away from the offline dataset under MLE training seen in Figure 1, Wu et al. [51] in practice augment the usual MLE loss with an entropy bonus as follows:

$$\pi_0 \doteq \pi_{\psi^\star} \quad \text{with} \quad \psi^\star \doteq \arg\max_\psi \left\{ \mathbb{E}_{(\mathbf{s}, \mathbf{a}) \sim \mathcal{D}} [\log \pi_\psi(\mathbf{a} \mid \mathbf{s}) + \beta \mathcal{H}(\pi_\psi(\cdot \mid \mathbf{s}))] \right\}. \quad \text{(A.12)}$$

where $\beta$ is temperature tuned to an entropy constraint similar to Haarnoja et al. [13]. The entropy bonus is estimated by sampling from the behavioral policy as

$$\mathcal{H}(\pi_\psi(\cdot \mid \mathbf{s})) = \mathbb{E}_{\mathbf{a} \sim \pi_\psi} [-\log \pi_\psi(\mathbf{a} \mid \mathbf{s})] \quad \text{(A.13)}$$

Figure 11 shows the predictive variances of behavioral policies trained on expert demonstrations for the "door-binary-v0" environment with various entropy coefficients $\beta$. Whilst entropy regularization partially mitigates the collapse of predictive variance away from the expert demonstrations, we still observe the wrong trend similar to Figure 1 with predictive variances high near the expert demonstrations and low on unseen data. The variance surface also becomes more poorly behaved, with "islands" of high predictive variance appearing away from the data.

We may also add Tikhonov regularization [12] to the MLE objective, explicitly,

$$\pi_0 \doteq \pi_{\psi^\star} \quad \text{with} \quad \psi^\star \doteq \arg\max_\psi \left\{ \mathbb{E}_{(\mathbf{s}, \mathbf{a}) \sim \mathcal{D}} [\log \pi_\psi(\mathbf{a} \mid \mathbf{s}) - \lambda \psi^\top \psi] \right\}. \quad \text{(A.14)}$$

where $\lambda$ is the regularization coefficient.

Figure 12 shows the predictive variances of behavioral policies trained on expert demonstrations for the "door-binary-v0" environment with varying Tikhonov regularization coefficients $\lambda$. Similarly, Tikhonov regularization does not resolve the issue with calibration of uncertainties. We also observe that too high a regularization strength causes the model to underfit to the variances of the data.

### A.4 Comparison to Prior Works

To assess the usefulness of KL regularization for improving the performance and sample efficiency of online learning with expert demonstrations, we compare our approach to methods that incorporate expert demonstrations into online learning implicitly or explicitly via KL regularization as well as by means other than KL regularization.

**ABM [44].** ABM explicitly KL-regularizes the online policy against a behavioral policy. This behavioral policy can be estimated via MLE, like BRAC, or alternatively via an "advantage-weighted behavioral model" where the RL algorithm is biased to choose actions that are both supported by the offline data and that are good for the current task. This objective filters trajectory snippets by advantage-weighting, using an $n$-step advantage function. We show that no carefully chosen objective with additional hyperparameters is required.

**AWAC [27].** AWAC performs online fine-tuning of a policy pre-trained on offline. It achieves state-of-the-art results on the dexterous hand manipulation and MuJoCo continuous locomotion tasks. AWAC implicitly constrains the KL divergence of the online policy to be close to the behavioral policy by sampling from the replay buffer, which is initially filled with the offline data. The method requires additional off-policy data to be generated to saturate the replay buffer, thereby requiring a hidden number of environment interactions that do not involve learning. Our approach does not require the offline data to be added to the replay buffer before training.

**AWR [31].** AWR approximates constrained policy search by alternating between supervised value function and policy regression steps. The objective derived is similar to AWAC but instead estimates the value function of the behavioral policy which was demonstrated to be less efficient than $Q$-function estimation via bootstrapping [27]. The method may be converted to use offline data by adding prior data to the replay buffer before training.

**BEAR [19].** BEAR attempts to stabilize learning from off-policy data (such as offline data) by tackling bootstrapping error from actions far from the training data. This is achieved by searching for policies with the same support as the training distribution. This approach is too restrictive for the problem considered in this paper, since only a small number of expert demonstrations is available, which requires exploration. In contrast, our approach encourages exploration away from the data by wider behavioral policy predictive variances. BEAR uses an alternate divergence measure to the KL divergence, Maximum Mean Discrepancy [45]. Other divergences such as Wasserstein Distances [30] have also been proposed for regularization in RL.

**BRAC [51].** BRAC regularizes the online policy against an offline behavioral policy as our method does. However, BRAC exhibits the pathologies we have shown by learning a poor behavioral policy via MLE. To mitigate this, in practice, BRAC adds an entropy bonus to the supervised learning objective which stabilizes the variance around the training set but has no guarantees away from the data. We demonstrate that behavioral policy obtained via maximum likelihood estimation with entropy regularization exhibit a collapse in predictive uncertainty estimates way from the training data, resulting in the pathology described in Proposition 1.

**DAPG [34].** DAPG incorporates offline data into policy gradients by initially pre-training with a behaviorally cloned policy and then augmenting the RL loss with a supervised-learning loss. We similarly pre-train the online policy at the start to avoid noisy KLs at the beginning of training. However, training a joint loss that combines two disparate and often divergent terms can be unstable.

**SAC+BC [26].** SAC+BC represents the approach of Nair et al. [26] but uses SAC instead of DDPG [22] as the underlying RL algorithm. The method maintains a secondary replay buffer filled with offline data that is sampled each update step, augmenting the policy loss with a supervised learning loss that is filtered by advantage and hindsight experience replay. Our method requires far fewer additional ad-hoc algorithmic design choices.

**SACfD [13].** SACfD uses the popular Soft Actor–Critic (SAC) algorithm with offline data loaded into the replay buffer before online training. Our algorithm uses the same approximate policy iteration scheme as SAC with a modified objective. Nair et al. [27] show that including the offline data into the replay buffer does not significantly improve the training performance over the unmodified SAC objective and that pre-training the online policy with offline data results in catastrophic forgetting. Thus, a different approach is needed to integrate offline data with SAC-style algorithms.

# Appendix B   Further Experimental Results

## B.1   Exploding $Q$-function Gradients

In Proposition 1 and Section 3.4, we showed that the policy gradient $\hat{\nabla}_\phi J_\pi(\phi)$ explodes due to the blow-up of the gradient of the behavioral reference policy's log-density as the behavioral policy predictive variance $\boldsymbol{\sigma}_0(\mathbf{s})$ tends to zero. A similar relationship holds for the $Q$-function gradients which we confirm empirically in Figure 8.

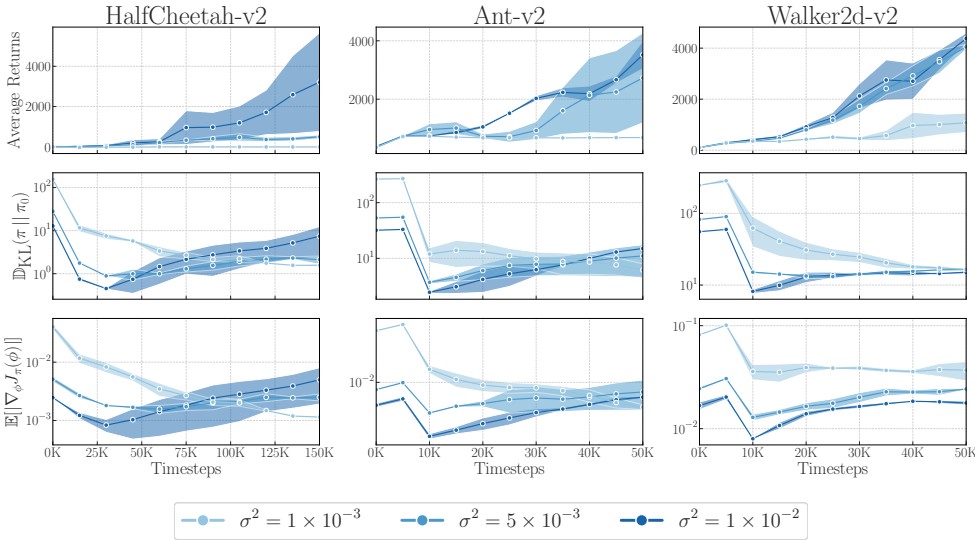

**Figure 8:** Ablation study showing the effect of predictive variance collapse on the performance of KL-regularized RL on MuJoCo benchmarks. Policies shown from dark to light in order of decreasing constant predictive variance, simulating training under maximum likelihood estimation. The plots show the average return of the learned policy, magnitude of the KL penalty, and magnitude of the $Q$-function gradients during online training.

## B.2   Ablation Study on the Effect of KL Divergence Temperature Tuning

Figure 9 shows that unlike in standard SAC [13], tuning of the KL-temperature is not necessary to achieve good online performance. For simplicity, we use a fixed value throughout our experiments.

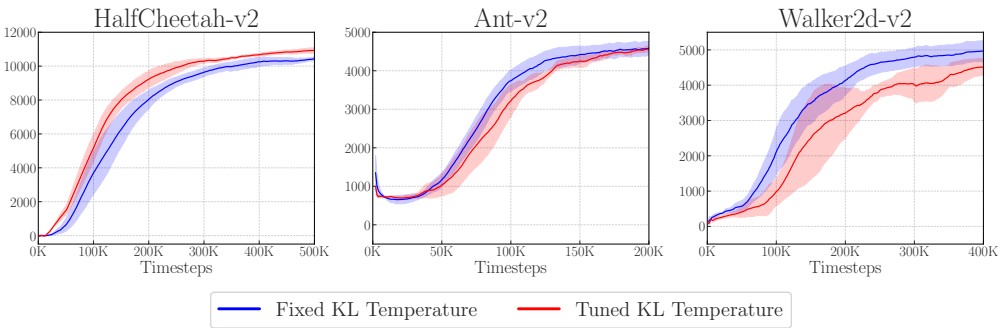

**Figure 9:** Ablation study on the effect of automatic KL temperature tuning on the performance of KL-regularized RL with a non-parametric GP behavioral reference policy on MuJoCo locomotion tasks.

## B.3 Ablation Study: Performance under a Laplace Parametric Behavioral Reference Policy

We use a Laplace behavioral reference policy to assess whether it is more effective at incorporating the expert demonstration data into online training. Figure 10 shows empirical results using the Laplace behavioral reference policy compared against N-PPAC (in blue) and a SAC baseline (in green) on three MuJoCo locomotion tasks. We use automatic KL-temperature tuning for this ablation. On the Ant-v2 environment, the Laplace behavioral reference policy slightly improves upon the baseline SAC performance, which does not use any prior information at all. On the door and pen environment, the online policy learned under the Laplace behavioral reference policy does not learn any meaningful behavior.

In both MuJoCo locomotion tasks and the "door-binary-v0" and "pen-binary-v0" dexterous hand manipulation environments, N-NPAC significantly outperforms both the online policy learned under the Laplace behavioral reference policy and the SAC baseline. We can understand the behavior under the Laplace behavioral reference policy in terms of collapse of predictive variance away from data for neural network parameterized policies, as it too has a decreasing variance away from the expert trajectories.

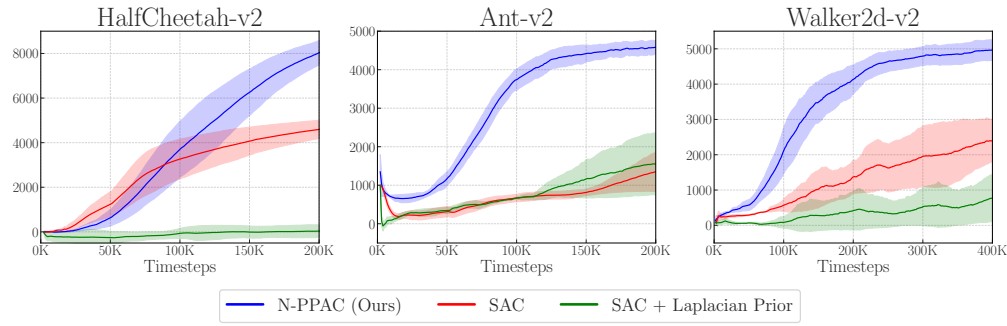

**Figure 10:** Ablation study using heavier-tailed Laplace behavioral reference policy on MuJoCo locomotion tasks.

**B.4   Visualizations of Regularized Maximum Likelihood Parametric Behavioral Policies**

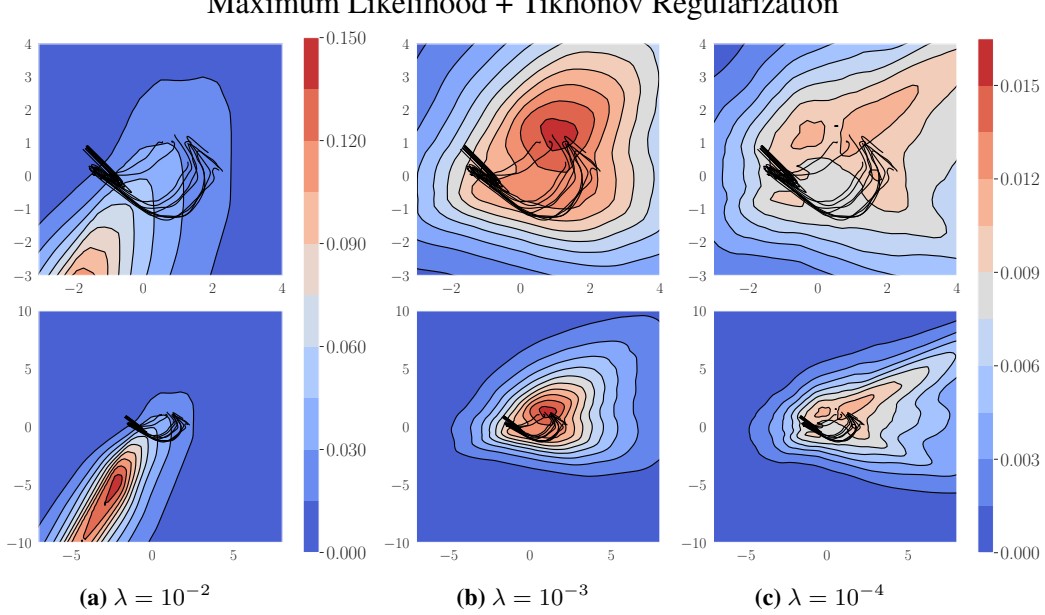

**Figure 11:** Predictive variances of parametric neural network Gaussian behavioral policies $\pi_\psi(\cdot \,|\, \mathbf{s}) = \mathcal{N}(\boldsymbol{\mu}_\psi(\mathbf{s}), \boldsymbol{\sigma}_\psi^2(\mathbf{s}))$ trained with different entropy regularization coefficients $\beta$.

**Figure 12:** Predictive variances of parametric neural network Gaussian behavioral policies $\pi_\psi(\cdot \,|\, \mathbf{s}) = \mathcal{N}(\boldsymbol{\mu}_\psi(\mathbf{s}), \boldsymbol{\sigma}_\psi^2(\mathbf{s}))$ trained with different Tikhonov regularization coefficients $\lambda$.

## B.5 Visualizations of Ensemble Maximum Likelihood Parametric Behavioral Policies

On the "door-binary-v0" environment, we consider an ensemble of parametric neural network Gaussian policies $\pi_{\psi^{1:K}}(\cdot \,|\, \mathbf{s}) \doteq \mathcal{N}(\boldsymbol{\mu}_{\psi^{1:K}}(\mathbf{s}), \boldsymbol{\sigma}_{\psi^{1:K}}^2(\mathbf{s}))$ with

$$\boldsymbol{\mu}_{\psi^{1:K}}(\mathbf{s}) \doteq \frac{1}{K}\sum_{k=1}^{K}\boldsymbol{\mu}_{\psi^k}(\mathbf{s}), \quad \boldsymbol{\sigma}_{\psi^{1:K}}^2(\mathbf{s}) \doteq \frac{1}{K}\sum_{k=1}^{K}\left(\boldsymbol{\sigma}_{\psi^k}^2(\mathbf{s}) + \boldsymbol{\mu}_{\psi^k}^2(\mathbf{s})\right) - \boldsymbol{\mu}_{\psi^{1:K}}^2(\mathbf{s}) \quad \text{(B.15)}$$

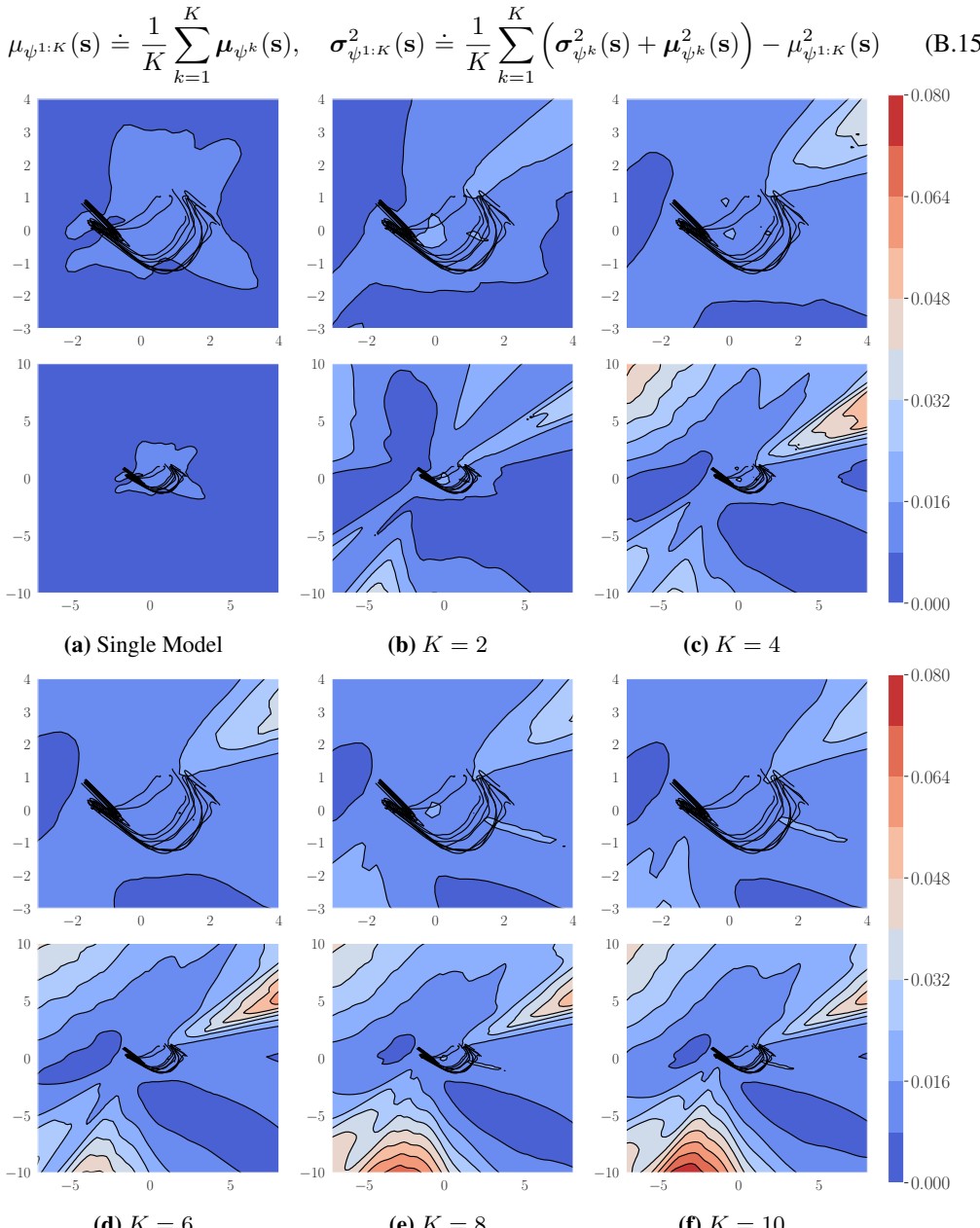

**Figure 13:** Predictive variances of ensembles of parametric neural network Gaussian behavioral policies $\pi_{\psi^{1:K}}(\cdot \,|\, \mathbf{s})$ with each neural network in the ensemble trained via MLE. The ensemble policies are marginally better calibrated than parametric neural network policies in that their predictive variance only collapses in some but not all regions away from the expert trajectories.

## B.6 Parametric vs. Non-Parametric Predictive Variance Visualizations Across Environments

Figure 14 shows the predictive variances of non-parametric and parametric behavioral policies on low dimensional representations of the environments considered in Figures 4 and 5 (excluding "door-binary-v0", which is shown in Figure 1).

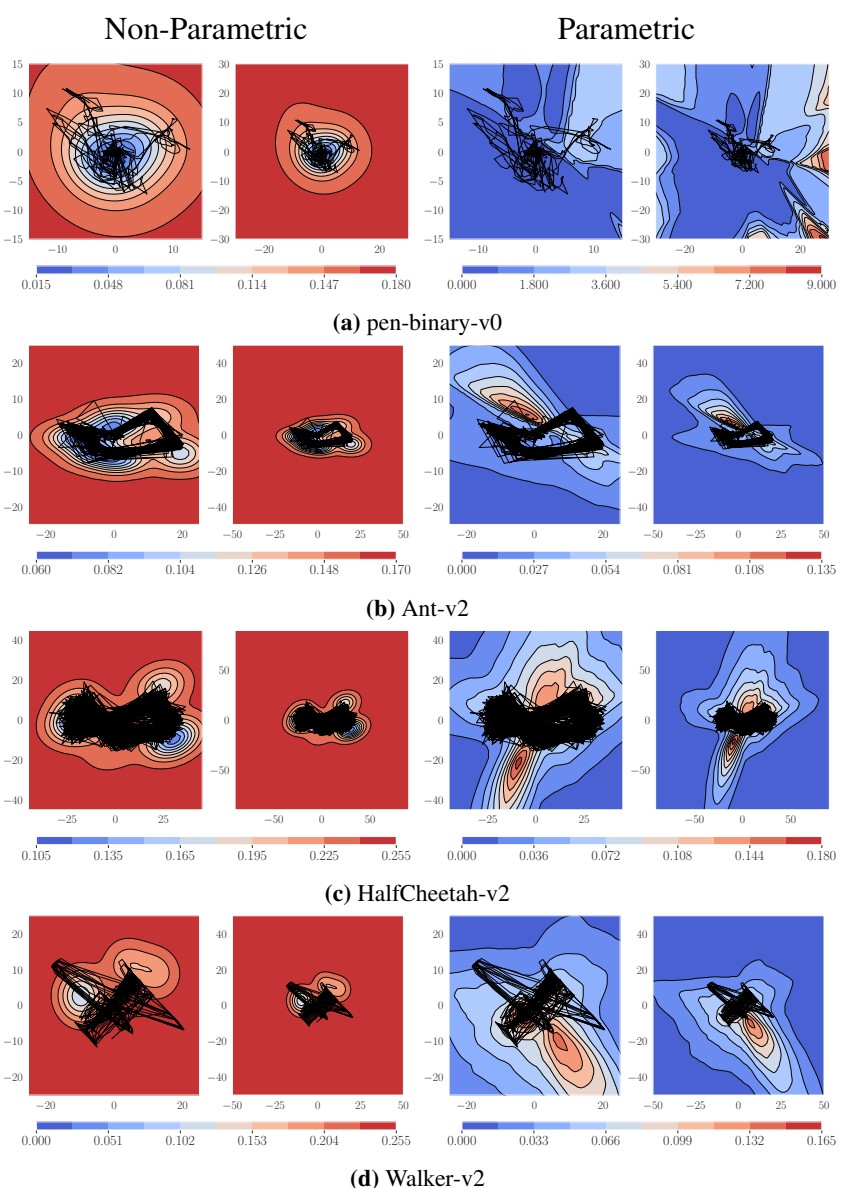

**Figure 14:** Predictive variances of non-parametric and parametric behavioral policies on low dimensional representations of the environments considered in Figures 4 and 5 (excluding "door-binary-v0", which is shown in Figure 1). **Left Column**: Non-parametric Gaussian process posterior behavioral policy $\pi_{\mathcal{GP}}(\cdot \mid \mathbf{s}, \mathcal{D}_0) = \mathcal{GP}(\boldsymbol{\mu}_0(\mathbf{s}), \boldsymbol{\Sigma}_0(\mathbf{s}, \mathbf{s}'))$. **Right Column**: Parametric neural network Gaussian behavioral policy $\pi_{\psi}(\cdot \mid \mathbf{s}) = \mathcal{N}(\boldsymbol{\mu}_{\psi}(\mathbf{s}), \boldsymbol{\sigma}_{\psi}^2(\mathbf{s}))$. Expert trajectories $\mathcal{D}$ used to train the behavioral policies are shown in black. As in Figure 1, the predictive variance of the GP is well-calibrated, whereas the predictive variance of the neural network is not.

## B.7 Visual Comparison of Parametric vs. Non-Parametric Behavioral Policy Trajectories

To better understand the significance of the behavioral policy's model class, we sample trajectories from different behavioral policies on the door-opening task in Figure 15. We visualize the mean trajectory and predictive variances of various behavioral policies showing a more sensible mean trajectory and predictive variance from the non-parametric GP policy leading to better regularization compared to a behavioral policy parameterized by a neural network and the implicit uniform prior in SAC, a state-of-the-art RL algorithm. On a randomly sampled unseen goal, we can see in Figure 15b that a neural network policy trained via MLE produces a confident but incorrect trajectory. The starting position is shown in black and the goal position is shown in green. We also visualize a uniform prior, which SAC implicitly regularizes against. Informative priors from offline data can greatly accelerate the online performance of such actor-critic methods.

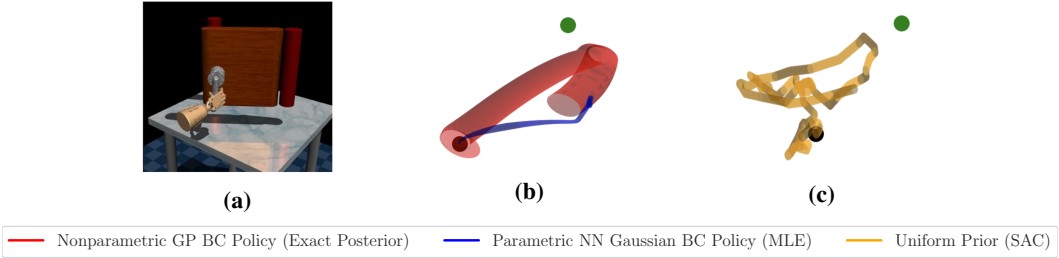

(a)  (b)  (c)

━━━ Nonparametric GP BC Policy (Exact Posterior)  ━━━ Parametric NN Gaussian BC Policy (MLE)  ━━━ Uniform Prior (SAC)

**Figure 15:** Left: challenging door opening task [35] which standard RL algorithms struggle on. Right and center: 3D plots of sampled mean trajectories and predictive variances from different behavioral policies from expert demonstration $\pi_0$, showing a more sensible mean trajectory and predictive variance from the non-parametric GP policy leading to better regularization over both: **(b)** a behavioral policy using a poor model class, and **(c)** the implicit uniform prior in SAC. Starting position shown in **black** and goal position shown in **green**.

# Appendix C  Further Implementation Details

## C.1  Algorithmic Details

**Pre-training** On the dexterous hand manipulation tasks, before online training, the online policy is pre-trained to minimize the KL divergence to the behavioral reference policy on the offline dataset:

$$J_{\mathcal{GP}}(\phi) \doteq \mathbb{E}_{\mathbf{s} \sim \mathcal{D}_0} \left[ \mathbb{D}_{\mathrm{KL}}(\pi_\phi(\cdot \mid \mathbf{s}) \parallel \pi_0(\cdot \mid \mathbf{s})) \right].$$

---

**Algorithm 1** Non-Parametric Prior Actor–Critic

---

**Input:** offline dataset $\mathcal{D}_0$, initial parameters $\theta_1$, $\theta_2$, $\phi$, GP $\pi_0(\cdot \mid \mathbf{s}) = \mathcal{GP}\big(m(\mathbf{s}), k(\mathbf{s}, \mathbf{s}')\big)$
Condition $\pi_0(\cdot \mid \mathbf{s})$ on $\mathcal{D}_0$ to obtain $\pi_0(\cdot \mid \mathbf{s}, \mathcal{D}_0)$
**for** each offline batch **do**
    $\phi \leftarrow \phi - \lambda_{\mathcal{GP}} \hat{\nabla}_\phi J_{\mathcal{GP}}(\phi)$     ▷ Minimize KL between online and behavioral reference policy.
**end for**

---

$\bar{\theta}_1 \leftarrow \theta_1, \bar{\theta}_2 \leftarrow \theta_2$     ▷ Initialize target network weights.
$\mathcal{D} \leftarrow \emptyset$     ▷ Initialize an empty replay pool.
**for** each iteration **do**
    **for** each environment step **do**
        $\mathbf{a}_t \sim \pi_\phi(\cdot \mid \mathbf{s}_t)$
        $\mathbf{s}_{t+1} \sim p(\cdot \mid \mathbf{s}_t, \mathbf{a}_t)$
        $\mathcal{D} \leftarrow \mathcal{D} \cup \big\{(\mathbf{s}_t, \mathbf{a}_t, r(\mathbf{s}_t, \mathbf{a}_t), \mathbf{s}_{t+1})\big\}$
    **end for**
    **for** each gradient step **do**
        $\theta_i \leftarrow \theta_i - \lambda_Q \hat{\nabla}_{\theta_i} J_Q(\theta_i)$ for $i \in \{1, 2\}$
        $\phi \leftarrow \phi - \lambda_\pi \hat{\nabla}_\phi J_\pi(\phi)$     ▷ Minimize $J_Q$ and $J_\pi$ using GP $\pi_0(\cdot \mid \mathbf{s}, \mathcal{D}_0)$.
        $\hat{\theta}_i \leftarrow \tau \theta_i + (1 - \tau)\hat{\theta}_i$ for $i \in \{1, 2\}$     ▷ Update target network weights.
    **end for**
**end for**
**Output:** Optimized parameters $\theta_1$, $\theta_2$, $\phi$

---

## C.2 Hyperparameters

Table 2 lists the hyperparameters used for N-PPAC. For other hyperparameter values, we used the default values in the RLkit repository. When multiple values are given, the former refer to MuJoCo continuous control and the latter to dexterous hand manipulation tasks.

**Table 2:** N-PPAC hyperparameters.

| Parameter | Value(s) |
|---|---|
| optimizer | Adam |
| learning rate | $3 \cdot 10^{-4}$ |
| discount ($\gamma$) | 0.99 |
| reward scale | 1 |
| replay buffer size | $10^6$ |
| number of hidden layers | $\{2, 4\}$ |
| number of hidden units per layer | 256 |
| number of samples per minibatch | $\{256, 1024\}$ |
| activation function | ReLU |
| target smoothing coefficient ($\tau$) | 0.005 |
| target update interval | 1 |
| number of policy pretraining epochs | 400 |
| GP covariance function | $\{$RBF, Matérn$\}$ |

Table 3 lists the hyperparameters used to train the Gaussian process on the offline data. The hyperparameters are trained by maximizing the log-marginal likelihood. The offline data is provided under the Apache License 2.0.

**Table 3:** GP optimization hyperparameters.

| Parameter | Value |
|---|---|
| optimizer | Adam |
| learning rate | 0.1 |
| number of epochs | 500 |

**Hyperparameter Sweep for Section 5.4.** For the BNN behavioral policy trained via Monte Carlo dropout, a dropout probability of $p = 0.1$ and a weight decay coefficient $1e - 6$ were used. These values were found via a hyperparameter search over $\{0.1, 0.2\}$ for $p$ and $\{1e-4, 1e-5, 1e-6, 1e-7\}$ for the dropout probability and the weight decay coefficient, respectively.

For the deep ensemble behavioral policy, $M = 15$ ensemble members and a weight decay coefficient of $1e - 6$ were used. The weight decay coefficient was found via a hyperparameter search over $\{5, 10, 15, 20\}$ for $M$ and $\{1e - 4, 1e - 5, 1e - 6, 1e - 7\}$ for the weight decay coefficient. Each ensemble member was trained on a different 80-20 training–validation split and initialized using different random seeds.