# OpenReview forum: "On Pathologies in KL-Regularized Reinforcement Learning from Expert Demonstrations"
_NeurIPS.cc/2021/Conference — NeurIPS 2021 Poster_

### Official Review · Reviewer_hW6R · 2021-07-05

**Rating:** 4
**Confidence:** 4

**Summary:**

The authors identify a potential failure in KL-regularized reinforcement learning in which small predictive variance of the target policy may lead to exploding gradients, possibly destabilizing gradient-based learning algorithms. The authors discuss an uncertainty collapse of parametric models under Maximum Likelihood estimation in OOD regions of the expert, resulting in the aforementioned pathology. Finally, they suggest to use non-parametric models of the behavior policy (in which uncertainty collapse is not present), showing improved performance on several continuous control benchmarks.

**Limitations And Societal Impact:**

The main limitation of the proposed approach is its scalability, which the authors discuss in Section 4.1 implicitly. I would be happy to see a more explicit discussion of this limitation.

**Main Review:**

The authors identify an interesting problem of KL regularized optimization and propose a solution to solve it. I find the problem interesting and the authors' solution to be well motivated and reasonable. In addition, the paper is written clearly, and the experiments were applied to reasonable tasks compared to similar / previous literature. The main limitations of the paper are as follows:

1. The model of the paper focuses on KL-regularized reinforcement learning. There are various modifications to this model which may trivially solve the authors' suggested pathology, yet the paper does not discuss them at all.
* First, KL-regularization is applied to the policies themselves and not the stationary distributions, d^\pi. Algorithms such as GAIL and AIRL can be combined with reward functions to construct regularized algorithms which utilize the expert data through stationary distributions. Would the issues the authors suggest still exist in this case? I would be interested to see comparison to such algorithms (with addition of reward).
* KL regularization is only one way to regularize the algorithm. Other f-divergences can be used which would not have the exploding gradient problem (e.g., TV-distance). Why do the authors focus on KL-regularized policies?

2. Scalability. While I agree that simple tasks would require small amounts of data, I find limited scalability as a major drawback of the paper. I hope for a solution that scales well with the amount of data provided, so that harder tasks could also be achieved by the proposed method.

3. Experimental Comparisons. As mentioned in 1 and 2 , the author's should compare their results to algorithms using stationary distributions (e.g., GAIL, AIRL), as well as other forms of f-divergences. I would also be happy to see comparison to CQL as well as DICE algorithms.

**Time Spent Reviewing:**

5

---

> ### Author Response · Authors · 2021-08-10
> **Response to Reviewer hW6R**
>
> **General comments**
>
> Thank you for your thoughtful review. We appreciated that you found the paper to be “written clearly” and the problem studied in the paper “interesting and [the proposed] solution to be well motivated and reasonable”.
>
> Below, we address your questions about our problem setting and choice of divergence metric as well as questions about the scalability of our method. We hope this response has addressed the concerns and questions raised in your review, but please let us know if any other issues remain.
>
> **1. “There are various modifications to this model which may trivially solve the authors' suggested pathology, yet the paper does not discuss them at all. [...] KL regularization is only one way to regularize the algorithm. Other f-divergences can be used which would not have the exploding gradient problem (e.g., TV-distance). Why do the authors focus on KL-regularized policies?”**
>
> We focus on KL-regularized RL, since it is widely used and has been studied extensively in the prior works [1,2,3,4]. There are indeed other, perhaps more appropriate, f-divergences, but KL-regularization is extremely popular and widely used in RL in practice (PPO, TRPO, etc.), making the study of potential pitfalls of KL-regularized RL particularly relevant. Given that KL-regularized RL is widely used in practice, we seek a solution to the pitfalls identified in our paper _without_ needing to change the underlying algorithm/optimization objective.
>
> More generally, thus far the interplay between the choice of model and KL-regularized objectives has not been studied widely, and we believe that is not yet sufficiently well-understood in the RL community. In this vein, our paper identifies a setting where prior work on KL-regularization did not recognize the pitfalls that could occur while using KL-regularized objectives, and it identifies a solution that does not require abandoning KL-regularized RL altogether, but only requires changing the model class of the behavioral policy. We hope that this work will encourage further study of the effect of different model classes on the efficacy of popular algorithms used in RL.
>
> Relatedly, we would like to point out that the behavior of (parametric) neural density model behavioral policies trained via Type I maximum likelihood, as shown in Figure 1 of the paper, is **undesirable regardless of the choice of divergence**. This is because for any choice of divergence, behavioral policies that exhibit high confidence (i.e., low predictive variance) on points far away from the expert demonstrations will heavily penalize the online policy for diverging from those poor mean predictions. While the resulting gradients may not increase quadratically for a given decrease in the predictive variance, for any choice of divergence, poor but high-confidence predictions will incentivize the online policy to learn a poor predictive mean on states away from the expert trajectory. Even if this doesn’t cause exploding gradients, with *any choice of divergence we do not want stronger regularisation towards points we are unsure about than points in our offline dataset*. Thus, we hypothesize that choosing a behavioral policy with better-calibrated predictive uncertainty would lead to a performance boost for all f-divergence choices.
>
> Lastly, we do want to point out that in the paper we do perform a comparison to methods that use other regularization schemes. While the focus of the paper is to fix the pathological behavior identified in KL-regularized RL, we show that KL-regularized RL **outperforms alternative methods that employ different regularization schemes**, such as BEAR with MMD, or SAC + BC and SACfD, with entropy regularization.
>
> **2. “KL-regularization is applied to the policies themselves and not the stationary distributions, d^\pi. Algorithms such as GAIL and AIRL can be combined with reward functions to construct regularized algorithms which utilize the expert data through stationary distributions. Would the issues the authors suggest still exist in this case? I would be interested to see comparison to such algorithms (with addition of reward).”**
>
> We stress that the GAIL and AIRL algorithms were designed for the inverse reinforcement learning setting, which is distinctly different from the setting we consider in our paper (i.e., online reinforcement learning augmented by offline expert demonstration state--action trajectories). Our work tackles a pathology in KL-regularized RL frequently encountered in prior work caused by interaction between the objective and model class. This type of interaction is of course not just restricted to policy regularization and may also arise when considering the stationary distributions. Furthermore, whilst the suggestion is good, we believe the empirical innovation required to be introduced to make that suggestion work belongs to another paper.
>
> **3. “Scalability. While I agree that simple tasks would require small amounts of data, I find limited scalability as a major drawback of the paper. I hope for a solution that scales well with the amount of data provided, so that harder tasks could also be achieved by the proposed method.”**
>
> We specifically investigate the expert demonstration setting, which typically considers small offline datasets. Indeed, as we show here [(anonymized link)](https://drive.google.com/file/d/1v16EuoTPY91LFS_ZnhXSZxbzXpR4Xe6E/view?usp=sharing) we can even further reduce the size of our dataset down to 1k points and still maintain the same performance. Nevertheless, for the two other behavioral policies investigated---using MC-Dropout BNN and deep ensembles---inference time does not scale with the number of training points and they do deliver performance stronger than the prior state-of-the-art on when trained on the full dataset.
>
> To provide a more precise evaluation of the GP vs. NN overhead, we isolated the time to compute the behavioral policy’s log-densities per epoch (1k env steps) used to estimate the KL divergence on two different environments: door-binary-v0 and HalfCheetah-v2. Each epoch involves 1000 mini-batches of size 256 and 1024, respectively, for door-binary-v0 and HalfCheetah-v2. The scaling of the GP depends both on the number of datapoints and the dimensionality of the action space. door-binary-v0 has an action space dimension of 28 and 5k expert demonstration samples, whereas HalfCheetah-v2 has an action space dimension of 6 and 15k expert demonstration samples.
>
> **Time to Compute KL Log-densities per Epoch (Single NN/GP Behavioral Policy)**
>
> | Dataset        | 1k data points | 5k datapoints  | 15k datapoints |
> |----------------|----------------|----------------|----------------|
> | HalfCheetah-v2 | 0.48s / 6.14s  | 0.47s / 12.42s | 0.48s / 55.72s |
> | door-binary-v0 | 0.59s / 12.99s | 0.59s / 46.92s |       -        |
>
> In the table above, we compare the overhead under different behavioral policies for expert demonstration data of varying size on two environments. The two numbers in each cell represent the time taken for a single neural density model behavioral policy (the first value) and for a GP behavioral policy (the second value). This shows for the experiments presented in the general response, we can maintain the performance of the GP behavioral policy on HalfCheetah-v2 while only incurring around 6s overhead from the KL computation per epoch. The base ‘per-epoch wall-clock time’ for SAC (which does not require computing the mean and variance of a behavioral policy) is around 20 seconds on a GeForce RTX 2080 GPU. Thus, **non-parametric GP behavioral policies require only approximately 30% additional wall-clock time per epoch but substantially fewer environment interactions**.
>
>
> **4. “I would also be happy to see comparison to CQL as well as DICE algorithms.”**
>
> We appreciate the suggestions for further prior benchmarks to accelerate online training from offline data. For the CQL/DICE algorithms, this has already been tried in Appendix F of the [AWAC](​​https://arxiv.org/abs/2006.09359) paper and has shown consistently worse performance than AWAC which we outperform on all the benchmarks that we consider.
>
> --
>
> References:
>
> [1] Nair, A., Dalal, M., Gupta, A. and Levine, S. (2020). Accelerating online reinforcement learning with offline datasets. arXiv preprint arXiv:2006.09359
>
> [2] Kumar, A., Fu, J., Soh, M., Tucker, G., and Levine, S. (2019). Stabilizing off-policy q-learning via bootstrapping error reduction. In Advances in Neural Information Processing Systems, pages 11761–11771.
>
> [3] Peng, X. B., Kumar, A., Zhang, G., and Levine, S. (2019). Advantage-weighted regression: Simple and scalable off-policy reinforcement learning. arXiv preprint arXiv:1910.00177.
>
> [4] Wu, Y., Tucker, G., and Nachum, O. (2019). Behavior regularized offline reinforcement learning. arXiv preprint arXiv:1911.11361.

---

> ### Author Response · Authors · 2021-08-26
> **Any remaining questions before the end of the discussion period?**
>
> Dear Reviewer hW6R,
>
> The discussion period ends on September 2 (in less than a week).
>
> We hope that our clarifications and the additional experimental results provided in our response addressed your questions and cleared up any misunderstandings about the problem setting and the comparisons included in the paper.
>
> Please let us know if you have remaining questions. We hope that we have addressed your questions and would appreciate it if you considered revising your score.
>
> Thank you.
>
> Sincerely,
> The Authors

---

### Official Review · Reviewer_Vbhn · 2021-07-16

**Rating:** 7
**Confidence:** 3

**Summary:**

This paper presents a previously unrecognized pathology in KL-regularized RL with expert demonstrations. Specifically the commonly used parametric behavioral policies suffer from collapse of predictive variance at states far from the demonstrations. This collapse of variance hinders the algorithm’s ability to learn effectively. The authors then propose to use non-parametric behavioral policies and demonstrate its effectiveness in several control tasks.


**Limitations And Societal Impact:**

The authors discussed the limitations of the proposed method and how this work could influence the research community to focus effort on better understanding of how policy classes would influence learning in deep RL. The authors did not explicitly comment on potential societal impact. One possible societal impact the authors could mention is that accurately estimating the predictive variance of ML systems is critical for real-world risk-sensitive applications such as autonomous driving.


**Main Review:**

Estimating predictive variance/uncertainty is an important topic that influences a broad range of ML research. Understanding and estimating the predictive variance in RL is in particular challenging due the instability of deep RL. This work provides an insightful analysis on KL-regularized RL algorithms and how the behavioral policy class influences the online learning process. The paper is well-written with clarity. The claims in the paper are supported with empirical experimental results and the designed experiments successfully highlight the problem of collapsed predicted variance at states not in demos in KL-regularized RL. This observation is novel and inspires the authors to propose a class of new algorithms that use non-parametric policies, which show comparable or better performance than prior methods in several high-dimensional control tasks.

It would be nice if the authors could also comment on how the non-parametric policy class would compare with parametric policies regularized with MC-dropout [1], since MC-dropout has also been considered as a way to do bayesian approximation on the network weights, and is often compared with Ensembles in related literature for estimating aleatoric uncertainty of neural network. In addition, in the paper the authors only commented on how the problem setting is friendly for non-parametric methods, it would be interesting to know what the actual computational trade-offs are, if any, for using non-parametric policies.

[1] Gal, Y., & Ghahramani, Z. (2016, June). Dropout as a bayesian approximation: Representing model uncertainty in deep learning. In international conference on machine learning (pp. 1050-1059). PMLR.



**Time Spent Reviewing:**

3

---

> ### Author Response · Authors · 2021-08-10
> **Response to Reviewer Vbhn**
>
> **General comments**
>
> Thank you for your positive and helpful review. We appreciated that you found the paper to be “well-written with clarity” and were pleased that you thought the analysis was “insightful” and that the “experiments successfully highlight the problem of collapsed predicted variance at states not in demos in KL-regularized RL.” We hope this response has addressed the concerns and questions raised in your review. Please let us know if any other issues remain.
>
>
> **1. “It would be nice if the authors could also comment on how the non-parametric policy class would compare with parametric policies regularized with MC-dropout”**
>
> That’s a good question. The reason we did not include BNNs in the paper’s empirical evaluation is that for BNN posterior predictive distributions (e.g., obtained via MC Dropout) we do not have access to the corresponding probability density function at a given evaluation point and hence cannot straightforwardly estimate the KL divergence or take gradients through it.
>
> One way to circumvent this issue is to consider a Gaussian behavioral policy whose moments are matched with the estimated predictive mean and variance of a given BNN. This way, we lose some flexibility in the BNN’s predictive distribution (because we assume it to be Gaussian), but are able to easily estimate the KL divergence in the policy objective. We conducted experiments with MC-Dropout BNN behavioral policies for this response and present the results in the general comment.
>
> We suspect that other variational inference methods for BNNs may lead to a better performance and hope to investigate this in more detail in follow-up work, but we will include results for MC-Dropout BNN behavioral policies on the full set of environments in the updated manuscript. In general, we expect BNN behavioral policies computed via variational inference---be it MC Dropout, mean-field variational inference (i.e., Bayes-by-Backprop), or other more recent methods---to underperform GPs, since they rely on approximate inference and are not guaranteed to have well-calibrated predictive uncertainty estimates, whereas non-parametric GP behavioral policies do.
>
>
> **2. “In addition, in the paper the authors only commented on how the problem setting is friendly for non-parametric methods, it would be interesting to know what the actual computational trade-offs are, if any, for using non-parametric policies.”**
>
> We would like to clarify that there is no inherent computational tradeoff between non-parametric and parametric models, but that there is a tradeoff between **exact** posterior inference in non-parametric models (which, for an appropriate choice of hyperparameters, is guaranteed to produce well-calibrated predictive uncertainty estimates and high predictive uncertainty estimates on data points ‘far away’ from the training data) on one hand and Type I maximum likelihood estimation in parametric models, such as Gaussian neural density models (which is likely to result in low predictive variance estimates on data points ‘far away’ from the training data), on the other hand. Exact inference in GPs scales cubically in the number of training points, whereas Type I maximum likelihood estimation in neural density models with mini-batch-based stochastic optimization has constant time complexity in the number of training data points.
>
> To provide a precise evaluation of the GP vs. NN overhead, we isolated the time to compute the behavioral policy’s log-densities per epoch (1k env steps) used to estimate the KL divergence on two different environments: door-binary-v0 and HalfCheetah-v2. Each epoch involves 1000 mini-batches of size 256 and 1024, respectively, for door-binary-v0 and HalfCheetah-v2. The scaling of the GP depends both on the number of datapoints and the dimensionality of the action space. door-binary-v0 has an action space dimension of 28 and 5k expert demonstration samples, whereas HalfCheetah-v2 has an action space dimension of 6 and 15k expert demonstration samples.
>
> **Time to Compute KL Log-densities per Epoch (Single NN/GP Behavioral Policy)**
>
> | Dataset        | 1k data points | 5k datapoints  | 15k datapoints |
> |----------------|----------------|----------------|----------------|
> | HalfCheetah-v2 | 0.48s / 6.14s  | 0.47s / 12.42s | 0.48s / 55.72s |
> | door-binary-v0 | 0.59s / 12.99s | 0.59s / 46.92s |       -        |
>
> In the table above, we compare the overhead under different behavioral policies for expert demonstration data of varying size on two environments. The two numbers in each cell represent the time taken for a single neural density model behavioral policy (the first value) and for a GP behavioral policy (the second value). This shows for the experiments presented in the general response, we can maintain the performance of the GP behavioral policy on HalfCheetah-v2 while only incurring around 6s overhead from the KL computation per epoch. The base ‘per-epoch wall-clock time’ for SAC (which does not require computing the mean and variance of a behavioral policy) is around 20 seconds on a GeForce RTX 2080 GPU. Thus, **non-parametric GP behavioral policies require only approximately 30% additional wall-clock time per epoch but substantially fewer environment interactions**.

---

### Official Review · Reviewer_vDkn · 2021-07-17

**Rating:** 7
**Confidence:** 3

**Summary:**

The authors identify a problem with KL-based reinforcement learning from expert demonstrations. Poor out-of-sample variance predictions of NN expert policies ends up unreasonably penalizing RL exploration outside the expert data distribution, potentially causing numerical instability. To remedy this they propose to instead use non-parametric expert policies based on a scalable GP approach.


**Limitations And Societal Impact:**

Adequate for this type of paper.

**Main Review:**

This seems like a solid incremental improvement. They thoroughly analyze an important problem in previous work and propose a fix. RL from expert demonstrations is a relevant niche with real-world applications. The paper is very well-written and the experimental results appear to improve on prior state-of-the-art.

Some comments:

- L43: "without ad-hoc design choices" - Like what? this is both needlessly vague and ends up sounding a bit objectionable.

- L178-185: I don't believe it is the limited capacity from its parameteric nature. Policies in RL are usually very simple and overparameterized. However, it is well-known that an NN trained with maximum likelihood will not predict out-of-distribution uncertainty well, simply because it hasn't been trained on that. Both the mean and variance predictions will be very poor outside the training distribution. It also sometimes goes to zero, but I'm not aware of a formal analysis of this, or convinced that it always happens. As you note in L200, ensembles are sometimes used for predictions, so they can't all always go to zero.

- Figure 6: Clarify which "Non-KL" methods are used in the bottom row, I assume it is AWAC etc? It might also have been nice to compare against regular RL with pre-trained (BC) policies as a sanity check.

- Section 5.3: Shouldn't you compare compute time against some RL algorithm *with* expert demonstrations? I understand that if it doesn't reach the reward thresholds without extra tuning, it becomes a bit apples to oranges, but this isn't ideal either. It's unclear how the overhead of the GP policy compares to the NN policy. It might be relevant to compare on the more complex benchmarks as well, as the simulator is usually the bottleneck in non-trivial RL applications.

**Time Spent Reviewing:**

4

---

> ### Author Response · Authors · 2021-08-10
> **Response to Reviewer vDkn**
>
> **General comments**
>
> Thank you for your positive and helpful review. We are pleased that you found the paper to be “very well-written” and recognize our work as a “solid incremental improvement” that “improve[s] on prior state-of-the-art.” We hope this response has addressed the concerns and questions raised in your review. Please let us know if any other issues remain.
>
>
> **1. “It's unclear how the overhead of the GP policy compares to the NN policy.”**
>
> To provide a more precise evaluation of the GP vs. NN overhead, we isolated the time to compute the behavioral policy’s log-densities per epoch (1k env steps) used to estimate the KL divergence on two different environments: door-binary-v0 and HalfCheetah-v2. Each epoch involves 1000 mini-batches of size 256 and 1024, respectively, for door-binary-v0 and HalfCheetah-v2. The scaling of the GP depends both on the number of datapoints and the dimensionality of the action space. door-binary-v0 has an action space dimension of 28 and 5k expert demonstration samples, whereas HalfCheetah-v2 has an action space dimension of 6 and 15k expert demonstration samples.
>
> **Time to Compute KL Log-densities per Epoch (Single NN/GP Behavioral Policy)**
>
> | Dataset        | 1k data points | 5k datapoints  | 15k datapoints |
> |----------------|----------------|----------------|----------------|
> | HalfCheetah-v2 | 0.48s / 6.14s  | 0.47s / 12.42s | 0.48s / 55.72s |
> | door-binary-v0 | 0.59s / 12.99s | 0.59s / 46.92s |       -        |
>
> In the table above, we compare the overhead under different behavioral policies for expert demonstration data of varying size on two environments. The two numbers in each cell represent the time taken for a single neural density model behavioral policy (the first value) and for a GP behavioral policy (the second value). This shows for the experiments presented in the general response, we can maintain the performance of the GP behavioral policy on HalfCheetah-v2 while only incurring around 6s overhead from the KL computation per epoch. The base ‘per-epoch wall-clock time’ for SAC (which does not require computing the mean and variance of a behavioral policy) is around 20 seconds on a GeForce RTX 2080 GPU. Thus, **non-parametric GP behavioral policies require only approximately 30% additional wall-clock time per epoch but substantially fewer environment interactions**.
>
>
> **2. “Shouldn't you compare compute time against some RL algorithm with expert demonstrations? I understand that if it doesn't reach the reward thresholds without extra tuning, it becomes a bit apples to oranges, but this isn't ideal either.”**
>
> We agree that the comparison in the paper should be changed as suggested by you.
>
> In the general response, we presented two reasonable alternatives to GP behavioral policies that may scale to larger expert demonstration datasets---MC-Dropout BNNs and deep ensembles---and found that, on the HalfCheetah-v2 environment, deep ensemble behavioral policies only slightly underperform GP behavioral policies and outperform (or perform on par with) the baselines included in the paper.
>
> To obtain the best online performance on this environment, for deep ensemble behavioral policies, we estimated the predictive mean and variance from 15 samples (i.e., using an ensemble consisting of 15 models and hence requiring 15 forward passes), and for MC-Dropout BNN behavioral policies, we estimated the predictive mean and variance from 20 MC samples (requiring 20 forward passes). Since the GP’s predictive mean and variance are computed analytically (and hence don’t need to be estimated from multiple forward passes), in the 1k data points case described in the general response, the time to compute the KL regularization per epoch is lower for a GP behavioral policy than for deep ensemble or MC dropout policies, since 0.48s x 15 = 7.2 (deep ensemble) > 6.14s (GP) and 0.48s x 20 = 9.6 (MC Dropout) > 6.14s, as can be seen from the table above.
>
>
> **3. “As you note in L200, ensembles are sometimes used for predictions, so they (the predictive variances of the individual Gaussian neural density models) can't all always go to zero.”**
>
> There may be a misunderstanding. The variance of a deep ensemble composed of Gaussian neural density models can be non-zero even if the predictive variance of each of its components is zero as long as the components’ predictive means differ. The is because the predictive variance of a mixture $\frac{1}{M} \sum_{m=1}^{M} \mathcal{N}\left(\mu_{\theta_{m}}(\mathbf{x}), \sigma_{\theta_{m}}^{2}(\mathbf{x})\right)$ (with $M$ ensemble components, each with weights $\theta_{m}$) is given by $\sigma_{\ast}^{2}(\mathbf{x})=\frac{1}{M} \sum_{m=1}^{M}\left(\sigma_{\theta_{m}}^{2}(\mathbf{x})+\mu_{\theta_{m}}^{2}(\mathbf{x})\right)-\mu_{*}^{2}(\mathbf{x})$ (also see Equation D.14 in our paper and Section 2.4 in Lakshminarayanan et al. (2017) [L2017]), which incorporates both disagreement in the components’ variance and their means. In our experiments, we consistently find that the predictive variance of all Gaussian neural density models trained on the expert data collapses to virtually zero on data points away from the expert trajectory.
>
>
> **4. “I don't believe it is the limited capacity from its parametric nature. [...] It is well-known that an NN trained with maximum likelihood will not predict out-of-distribution uncertainty well, [...] but I'm not aware of a formal analysis of this, or convinced that it always happens.”**
>
> We generally agree that neural density models (such as a Gaussian distribution with mean and variance parameterized by a NN) trained via Type I maximum likelihood are not expected to perform well on out-of-distribution inputs and that this behavior is well-known and apart from the discussion in Quinonero-Candela and Rasmussen (2005) [QR2005] referenced in the manuscript, we too are not aware of a formal analysis of this specific setting.
>
> Quinonero-Candela and Rasmussen (2005) discuss the effect of degeneracy in probabilistic models and state that a degenerate (i.e., parametric) distribution over “functions [can be] so restrictive, that given enough data only a very limited family of functions will be plausible under the posterior, leading to overconfident predictive variances.” While the parametric NN behavioral policy distributions in the paper differ from the models considered in Quinonero-Candela and Rasmussen (2005), a similar effect may apply here. Rasmussen & Williams (2005) [RW2005] further note that the capacity of non-parametric models increases with more data, while it stays constant for parametric models.
>
>
> **5. “It might also have been nice to compare against regular RL with pre-trained (BC) policies as a sanity check.”**
>
> “SAC + BC” in the empirical evaluation does exactly that using the SAC algorithm for online training. As we show in Figures 4 and 6, this approach works relatively well on MuJoCo environments, but performs poorly on the more challenging pen-binary-v0 and door-binary-v0 environments.
>
>
> **6. “Figure 6: Clarify which "Non-KL" methods are used in the bottom row, I assume it is AWAC etc?”**
>
> SAC + BC, SACfD (SAC from demonstrations), BEAR, and DAPG, do not use KL regularization against a behavioral policy. AWAC implicitly uses KL-regularization. We provide a detailed description of each algorithm in Appendix H and will make this clearer in the main text.
>
>
> **7. “‘without ad-hoc design choices’ - Like what? this is both needlessly vague and ends up sounding a bit objectionable.”**
>
> We appreciate the pointer. From the paper and [publicly available code](https://github.com/rail-berkeley/rlkit/blob/6a13e1b63d3febb1057f46e6bb6b86948b9cab1e/examples/awac/hand/awac1.py#L27) of related works (e.g., AWAC), we are aware that prior work dealt with instabilities related to the variance collapse in an ad-hoc manner, for example, by setting the variance to a constant value for all states or bounding it away from zero. We agree that the statement is not sufficiently specific and will remove it from the abstract. We will instead include a short discussion of the specific ad-hoc choices used in prior works in the main paper.
>
>
> --
>
> References:
>
> [L2017] Balaji Lakshminarayanan, Alexander Pritzel, Charles Blundell.
> Simple and Scalable Predictive Uncertainty Estimation using Deep Ensembles. Advances in Neural Information Processing Systems 30, 2017.
>
> [QR2005] Joaquin Quinonero-Candela, Carl Edward Rasmussen. A Unifying View of Sparse Approximate Gaussian Process Regression. Journal of Machine Learning Research, 2005.
>
> [RW2005] Carl Edward Rasmussen and Christopher K. I. Williams. Gaussian Processes for Machine Learning. MIT Press, 2005.

---

> > ### Comment · Reviewer_vDkn · 2021-09-05
> > **Maintaining my score.**
> >
> > I thank the authors for their candid responses. This allays most of the concerns I had. I will maintain my score.

---

> > > ### Author Response · Authors · 2021-09-05
> > > **Thank you for your feedback!**
> > >
> > > Thank you for taking the time to read and reply to our response. We appreciate your feedback and your contribution to the discussion. We will include the clarifications and additional experiments from our response in the updated draft.

---

### Official Review · Reviewer_3qqH · 2021-07-23

**Rating:** 7
**Confidence:** 5

**Summary:**

This paper studies the problem of KL-regularized reinforcement learning for locomotion tasks, where the KL regularization penalizes deviations from expert demonstrations.

This work makes the observation that fitting expert demonstrations to a certain class of conditional neural density models (e.g. neural networks that output the parameters of a Gaussian distribution), results in policies whose variance collapses for states that are different enough from the expert data.

The paper argues that this collapse causes instabilities in learning KL-regularized policies: for Gaussian policies, the KL penalty grows quadratically to infinity as the expert policy variance goes to 0. To address this issue, the paper proposes to instead compute the KL penalty by fitting expert demonstrations to models that do not suffer from this collapse, in particular Gaussian Process regression models. Gaussian Process models result in variance that increases (depending on the choice of kernel) for states that are far enough from the expert data.

The paper provides some analysis on why the variance collapse affects the RL optimization process, experiments that  aim to show an empirical relationship between expert policy variance, KL penalty and magnitude of the policy gradients, and a comparison between various methods that use KL regularization with an expert policy.


**Limitations And Societal Impact:**

Section 4.1, which the authors point to as a description of the model limitations at the end of the paper, only mentions the cubic complexity of fitting GP models with exact inference. Further, this limitation is presented as not really being a limitation due to existing algorithms based on conjugate gradient solvers for exact inference with GPs. The paper, however, fails to mention other important limitations, in particular ones affecting the current implementation: how well will the method work with non-smooth data? with high dimensional input data such as images? How can the model be extended to multimodal expert behaviours? While the paper does not need to provide solutions to these issues, the discussion of limitations would be more helpful if it pointed out problems that are not easily addressable.


**Main Review:**

As summarized above, this paper presents a method for KL regularized RL, in which the KL penalty computation uses a behavioural policy prior fit from expert data. The proposed method uses Gaussian Process models for fitting the expert data. The paper is clearly written, proposes a simple approach and provides empirical evidence of its usefulness.

It is notable that, in the experiments presented in the paper, the resulting learned policies produce visually smooth behaviours when regularized with a smooth prior (even if the policy class is not smooth). The paper analyses the problem of variance collapse in KL regularized RL, showing the relationship between variance collapse (which is empirically observed) with optimization instabilities due to exploding gradients.

Some things to improve:

The paper limits its comparison to one between conditional Gaussian models with ReLU networks and GP models with smooth stationary kernels. While this is enough for the main point of the paper (addressing the variance collapse), it doesn't necessarily mean that parametric models will have variance that will invariably collapse, particularly in the case where modelling uncertainty is introduced in the parametric model fitting (e.g. the comparison with ensembles). The paper does not need to answer how parametric models could be made to work in this setting, but the main text should be explicit about what is specifically being compared.

An useful comparison that is missing in this story is the effect of expert dataset size on the performance of the algorithms. Is it always better to use GP regression for fitting the prior as the expert dataset size changes?

What kind of ensemble was used for the experiments (bootstrap, different random seeds/initialization)? Since the paper introduces a comparison with ensembles, another useful comparison of the method would be against an ensemble model, as you let the size of the ensemble grow.  In the context of the discussion about ensembles, it would be helpful to include a discussion about other models that could be applied in this setting (e.g. bayesian neural networks). and their limitations.

A comment on the robotics experiments: in general, human expert demonstrations are not readily available for robot tasks. Collecting human expert data is still expensive and Perhaps the authors meant that there exists publicly available datasets for learning from demonstrations for robot manipulation tasks.


**Time Spent Reviewing:**

6

---

> ### Author Response · Authors · 2021-08-10
> **Response to Reviewer 3qqH**
>
> **General comments**
>
> Thank you for your detailed and attentive review. We appreciate your thoughtful questions and were happy you found the paper to be “clearly written”. We hope this response has addressed the concerns and questions raised in your review. Please let us know if any other issues remain.
>
>
> **1. “It would be helpful to include a discussion about other models that could be applied in this setting (e.g. bayesian neural networks) and their limitations.”**
>
> That’s a good suggestion. The reason we did not include BNNs in the empirical evaluation in the paper is that for BNN posterior predictive distributions (e.g., obtained via MC Dropout), we do not have access to the corresponding probability density function and hence cannot straightforwardly estimate the KL divergence or take gradients through it.
>
> One way to circumvent this issue is to consider a Gaussian behavioral policy whose moments are matched with the estimated predictive mean and variance of a given BNN. This way, we lose some flexibility in the BNN’s predictive distribution (because we assume it to be Gaussian), but are able to easily estimate the KL divergence in the policy objective. We conducted experiments with MC-Dropout BNN behavioral policies for this response and present the results in the general comment.
>
> We suspect that other variational inference methods for BNNs may lead to a better performance and hope to investigate this in more detail in follow-up work, but we will include results for MC-Dropout BNN behavioral policies on the full set of environments in the updated manuscript. In general, we expect BNN behavioral policies computed via variational inference---be it MC Dropout, mean-field variational inference (i.e., Bayes-by-Backprop), or other more recent methods---to underperform GPs, since they rely on approximate inference and are not guaranteed to have well-calibrated predictive uncertainty estimates, whereas non-parametric GP behavioral policies do.
>
>
> **2. “Is it always better to use GP regression for fitting the prior as the expert dataset size changes?”**
>
> That’s a good question. GPs are known to be particularly well-suited to small-dataset-size regimes (Rasmussen & Williams (2005) [RW2005]), so we would expect the performance of the GP relative to the parametric models to improve for smaller offline datasets (i.e., fewer expert demonstrations). Our experiments in the general response confirm this hypothesis with GPs noticeable outperforming MC-Dropout BNNs and Deep Ensembles when given only a single trajectory of 1k points on the HalfCheetah-v2 environment.
>
>
> **3. “What kind of ensemble was used for the experiments (bootstrap, different random seeds/initialization)?”**
>
> For the experiments presented in the paper, we used an ensemble created from multiple neural density models, each trained using a different seed/initialization but with the same training dataset.
>
> For this rebuttal, we re-ran our experiments on the door-binary-v0 environment and additionally conducted new experiments (described in the general response) on the HalfCheetah-v2 environment with an ensemble behavioral policy (using bootstrapping and different seeds/initializations).
>
>
> **4. “Since the paper introduces a comparison with ensembles, another useful comparison of the method would be against an ensemble model, as you let the size of the ensemble grow.”**
>
> We have now considered a deep (bootstrap) ensemble with different initialisations for the experiments in the general response. We evaluated ensembles with member count $M$ in {5, 10, 15, 20} and found that $M=15$ performed the best.
>
>
> **5. “It doesn't necessarily mean that parametric models will have variance that will invariably collapse, particularly in the case where modelling uncertainty is introduced in the parametric model fitting (e.g. the comparison with ensembles)”**
>
> We agree. To test whether parametric models that are able to quantify their predictive uncertainty perform better than the deterministic NNs, we included ensemble behavioral policies in our comparison on the door-binary-v0 environment (see Figure 6, right) and showed that they underperform GP behavioral policies. We also plotted the predictive uncertainty of an ensemble behavioral policy in Figure 10 (in the appendix) and show that it is better calibrated than that of parametric NN policies in the sense that it only collapses in some but not all regions away from the expert trajectories.
>
> For this rebuttal, we conducted additional experiments with deep ensemble behavioral policies on HalfCheetah-v2 and find that online training with ensemble behavioral policies (using bootstrapping and different seeds/initializations) outperforms online training with single NN behavioral policies but **still underperforms online training with GP behavioral policies**. We will emphasize this result in the updated manuscript to highlight deep ensemble behavioral policies as a worse but scalable alternative.
>
>
> **6. “The paper limits its comparison to one between conditional Gaussian models with ReLU networks and GP models with smooth stationary kernels.”**
>
> There may be a misunderstanding. In our hyperparameter search, we used GPs with both smooth (RBF) and rough (Matern-$\frac{1}{2}$) kernels, both of which outperformed conditional Gaussian NN models, but found RBF kernels to be better for MuJoCo and Matern-$\frac{1}{2}$ kernels to be better for the dexterous hand manipulation tasks. This is listed in Appendix J but we will make this more clear in the main paper. Thank you for the pointer!
>
>
> **7. “the resulting learned policies produce visually smooth behaviours when regularized with a smooth prior (even if the policy class is not smooth)”**
>
> There may be a misunderstanding about Figure 1 in the main paper (and Figures 8-11) in the appendix. The figures show the predictive variance of different behavioral policies at different points in (a low-dimensional representation of the) state space. In fact, the functions $\pi_{0}(a | s)$ are non-smooth both for parametric (neural network) and non-parametric (GP) approximators, since we use ReLU activations for the neural networks and Matern-$\frac{1}{2}$ kernels for the GP on “door-binary-v0”. We evaluated the GP performance on both RBF and Matern-$\frac{1}{2}$ kernels (see Table 2 in the appendix for the hyperparameters used), but found that Matern-$\frac{1}{2}$ resulted in a slightly better performance for the hand manipulation tasks. We will indicate this more clearly in the manuscript.
>
>
> --
>
> References:
>
> [RW2005] Carl Edward Rasmussen and Christopher K. I. Williams. Gaussian Processes for Machine Learning. MIT Press, 2005.

---

### Author Response · Authors · 2021-08-10
**General Comments & Alternative Parametric Behavioral Policies**

**General Comments**

We thank all reviewers for the detailed and thoughtful feedback. We are excited about the new conference guidelines encouraging authors and reviewers to engage in an active discussion over the next two weeks and look forward to the exchange. We hope this will help resolve any remaining misunderstandings in the reviews or in our response. We will first make a general comment on common questions and then refer to individual responses where we address specific comments and further clarify ambiguities in the reviews.

We were pleased that reviewers noted our paper “analysed an important problem in previous work” and presented a “well-motivated” solution along with empirical evidence that “it improved on prior state-of-the-art”.

We greatly appreciated your questions and comments, which will greatly improve the updated paper.

**Alternative Parametric Behavioral Policies**

A common theme amongst the reviewers was raising the possibility that alternate classes of behavioral policies such as MC-Dropout BNNs and Deep Ensembles may also go some way to resolving the pathology identified in our paper. A related question raised was on how useful GP behavioral policies are when only very few expert demonstrations are available, which is relevant as expert demonstrations are often limited in number.

To address these questions, below, we present empirical results on the HalfCheetah-v2 dataset (15k in size) on which we compare several behavioral policies---non-parametric GPs, deep ensembles, and MC-Dropout BNNs---on both the full dataset of 15 expert demonstrations (15 state--action trajectories with a total of 15k samples) and on only a single expert demonstration (a single state--action trajectory with a total of of 1k samples) with the goal of assessing the usefulness of the different behavioral policies under varying levels of expert data availability.

We show the results for the full dataset here [(anonymized link)](https://drive.google.com/file/d/15RbYtsO0Pk5UQThPx-PFpLjSsWMu1To6/view?usp=sharing) and for a single trajectory here [(anonymized link)](https://drive.google.com/file/d/1v16EuoTPY91LFS_ZnhXSZxbzXpR4Xe6E/view?usp=sharing). On the full dataset, we find all three methods are competitive and improve the prior SOTA but that the GP behavioral policy leads to the highest return. The ensemble behavioral policy leads to the second highest return, and the MC Dropout behavioral policy leads to the lowest return.

Remarkably, on the single trajectory reduced dataset, we find that **GPs perform just as well with only a single expert trajectory as with all 15 (i.e., with 1k data points, vs. 15k data points)**. The deep ensemble (using bootstrapping and different seeds/initializations) still performs remarkably well, but the MC-Dropout BNN performs poorly.

Both the MC-Dropout BNN and deep ensemble’s hyperparameters were extensively tuned.
This further emphasizes the usefulness of non-parametric behavioral policies when accelerating online training with expert demonstrations---even when only very little expert demonstration data is available. We will include these empirical evaluations in the main paper and aim to perform the same analysis for the other environments as well.

Lastly, we note the significance of these results for the scalability of the proposed fix to the pathological behavior identified in the paper.

(i) On one hand, the results demonstrate that even if a surprisingly large amount of expert data is provided, when using a non-parametric GP behavioral policy, it may be possible to only use a sufficiently small subset of the expert demonstrations (roughly <20k samples) and still obtain the same final performance as with a much larger expert demonstration data. Of course, whether this is possible will likely depend on the environment and the task.

(ii) On the other hand, the results also demonstrate that deep ensembles (with bootstrapping and different seeds/initializations) and BNNs provide simple and easily scalable alternative behavioral policies for cases where a surprisingly large amount of expert data is provided. While our results show that both models slightly underperform the GP behavioral policy on the HalfCheetah-v2 environment, they both outperform single neural density models trained via Type I maximum likelihood. Although we were unable to achieve above-zero success rates on the more challenging door-binary-v0 environment using deep ensembles and MC-Dropout BNN behavioral policies, our experiments suggest that they may be useful alternatives to non-parametric GP behavioral policies for some environments.

Implementation details:

For the MC-Dropout BNN behavioral policy, we used a dropout probability of $p=0.1$ and a weight decay coefficient $1e-6$, which we found via a hyperparameter search over {$0.1, 0.2$} for $p$ and {$1e-4, 1e-5, 1e-6, 1e-7$} for the dropout probability and the weight decay coefficient, respectively.

For the deep ensemble behavioral policy, we used $M=15$ members and a weight decay coefficient of $1e-6$. We found these via a hyperparameter search over {$5, 10, 15, 20$} for $M$ and {$1e-4, 1e-5, 1e-6, 1e-7$} for the weight decay coefficient. Each ensemble member used a different 80-20 test--validation split and was also initialized differently.

---

### Author Response · Authors · 2021-09-04
**Summary of Discussion**

Dear all,

We’d like to again thank you all for your time, and hope our responses helped clarify any remaining questions you may have had.

In case you find it helpful, we summarized the main points from the reviews (as we understood them) and our responses below:

**Could  the pathology related to KL-penalties be “trivially” avoided?**

Reviewer hW6R remarked that “there are various modifications to this model which may trivially solve the suggested pathology, yet the paper does not discuss them at all.” The reviewer made two suggestions, both of which boil down to changing the problem setting (i.e., not using KL-regularization on policies in the first place), rather than fixing the issue of KL-regularized RL discussed in the paper. KL-regularized objectives in RL are well-established, easy to compute, and widely used (especially for online RL with expert demonstrations and offline RL), and identifying and addressing problems specific to KL-regularized RL is an important research direction.

We addressed the two suggestions made by the reviewer in our response. A short summary:

- We explained that our empirical evaluation already includes some of the “trivial” fixes suggested (e.g., changing the KL divergence to a different distance metric, such as the maximum mean discrepancy (MMD) or the policy entropy). Our evaluation showed that these fixes can improve performance somewhat, but still perform worse than KL-regularized RL combined with the remedy proposed in our paper. This shows that---irrespective of the regularization method used---behavioral policies that do not assign high predictive variance away from the data (as shown in Figure 1) will lead to a degradation in performance. Changing the model class of the behavioral policy as suggested in our paper may therefore be useful for other regularization methods as well (although this is not the focus of our paper).

- We explained that the suggestion regarding the use of KL divergences between different distributions (for example, state--action occupancy measures) may be worthwhile to pursue, but that the specific suggestion made by the reviewer referred to the setting of inverse RL, which is distinctly different from the “online learning from expert demonstration” considered in our submission. The suggestion may be useful, but outside of the scope of the paper and does not directly fit the problem setting considered in it.


**Is it necessary to use a GP behavioral policy or would BNNs/ensembles suffice to address the pathology?**

In our response, we presented additional results with an MC-dropout BNN and a deep (bootstrapped) ensemble on the full dataset (15k) and a reduced dataset (1k, a single trajectory). We provided a detailed discussion of the results in the [general comment](https://openreview.net/forum?id=sS8rRmgAatA&noteId=JMlXKEMP7E).

A short summary of the main take-aways:

- For the full half-cheetah dataset, the deep ensemble can perform as well as the GP but the MC-dropout BNN will perform worse. For the reduced dataset, both the deep ensemble and MC-dropout BNN perform significantly worse than the GP, demonstrating the strength of GP models especially for learning from expert demonstrations, which are typically small in number.

- For the more challenging door-binary-v0 environment, neither the deep ensemble nor the MC-dropout BNN provided effective regularization to the online policy. This is consistent with previous poor results from parametric policies as with BRAC. Partial explanation for this can be given by Figure 10 in the Appendix, which shows that ensemble behavioral policies do not produce well-calibrated uncertainty estimates compared to the GP.

- We will include all of these results in the updated paper.

We hope that our response has answered all of your questions and addressed any potential concerns you may have had. Thank you for your time and feedback on our submission! Please don't hesitate to let us know if you have any remaining questions or concerns.

Sincerely,

The authors

---

### Decision · Program_Chairs · 2021-09-27

**Decision:**

Accept (Poster)

**Comment:**

There has been a split amongst the reviewers for this paper. Although each expressed interest in the pathology of KL-regularization stated by the author, there has not been a consensus about whether the author's approach to this is impactful. I would recommend the position of the majority, 3 reviewers have found the set of baselines and architectures attempted are suitable to explore the problem at hand, and the emirical results are promising enough to be of value to the community as written (and with promised helpful additions in the discussion). Therefore, I recommend acceptance.

I do also sympathize with the position of reviewer hW6R, in that I may not be likely to use KL-regularization to solve the RL-from expert problem as of this year and have concerns about scaling GPs for high-dimensional problems. While these concerns are relevant, and could inform the authors about helpful discussion in the final version or their next papers, I still believe the contribution of a focused analysis of this widely used form of KL-regularization is sufficient in current form.